Primates in peril: the significance of Brazil, Madagascar, Indonesia and the Democratic Republic of the Congo for global primate conservation

Estrada Alejandro 1 aestradaprimates@gmail.com
http://orcid.org/0000-0003-0053-8356 Garber Paul A. 2 p-garber@illinois.edu
Mittermeier Russell A. 3
Wich Serge 4
Gouveia Sidney 5
Dobrovolski Ricardo 6
http://orcid.org/0000-0001-5523-7353 Nekaris K.A.I. 7
Nijman Vincent 7
Rylands Anthony B. 3
http://orcid.org/0000-0002-0778-0615 Maisels Fiona 8 9
http://orcid.org/0000-0001-6848-9154 Williamson Elizabeth A. 9
http://orcid.org/0000-0002-5400-845X Bicca-Marques Julio 10
http://orcid.org/0000-0003-0955-8214 Fuentes Agustin 11
http://orcid.org/0000-0003-0744-1987 Jerusalinsky Leandro 12
Johnson Steig 13
Rodrigues de Melo Fabiano 14
http://orcid.org/0000-0002-1774-0713 Oliveira Leonardo 15
Schwitzer Christoph 16
Roos Christian 17
Cheyne Susan M. 18 19
Martins Kierulff Maria Cecilia 20
Raharivololona Brigitte 21
http://orcid.org/0000-0001-6783-2715 Talebi Mauricio 22
Ratsimbazafy Jonah 23
Supriatna Jatna 24
Boonratana Ramesh 25
Wedana Made 26
Setiawan Arif 27
1 Institute of Biology, National Autonomous University of Mexico (UNAM) , Mexico City , Mexico
2 Department of Anthropology, University of Illinois at Urbana-Champaign , Urbana, IL , USA
3 Global Wildlife Conservation , Austin, TX , USA
4 School of Natural Sciences and Psychology and Institute for Biodiversity and Ecosystem Dynamics, Liverpool John Moores University and University of Amsterdam , Liverpool , UK
5 Department of Ecology, Federal University of Sergipe , São Cristóvão , Brazil
6 Department of Zoology, Federal University of Bahia , Salvador , Brazil
7 Department of Social Sciences, Oxford Brookes University , Oxford , UK
8 Global Conservation Program, Wildlife Conservation Society , NY , USA
9 Faculty of Natural Sciences, University of Stirling , Stirling, Scotland , UK
10 Pontifícia Universidade Católica do Rio Grande do Sul , Porto Alegre , Brazil
11 Department of Anthropology, University of Notre Dame , Notre Dame, IN , USA
12 Instituto Chico Mendes de Conservação da Biodiversidade, Ministério do Meio Ambiente , Brasilia , Brazil
13 Department of Anthropology and Archaeology, University of Calgary , Calgary, AB , Canada
14 Universidade Federal de Goiás and Dept. Eng. Florestal, Campus UFV, UFV, Viçosa, Brazil , Jataí Viçosa , Brazil
15 Departamento de Ciências, Faculdade de Formação de Professores, Universidade do Estado do Rio de Janeiro (DCIEN/FFP/UERJ) , Rio de Janeiro , Brazil
16 Bristol Zoological Society , Bristol , UK
17 Deutsches Primatenzentrum, Leibniz Institute for Primate Research , Göttingen , Germany
18 Borneo Nature Foundation , Palangka Raya , Indonesia
19 Oxford Brookes University , Oxford , UK
20 Universidade Federal do Espírito Santo, Instituto Pri-Matas and Centro Universitário Norte do Espírito Santo , Belo Horizonte , Brazil
21 Mention Anthropobiologie et Développement Durable, University of Antananarivo , Antananarivo , Madagascar
22 Universidade Federal de São Paulo , Diadema, São Paulo , Brazil
23 Groupe d’étude et de recherche sur les primates (Gerp) , Antananarivo , Madagascar
24 Graduate Program in Conservation Biology, Department of Biology FMIPA, University of Indonesia , Depok , Indonesia
25 Mahidol University International College , Salaya, Nakhon Pathom , Thailand
26 The Aspinall Foundation–Indonesia Program , Bandung West Java , Indonesia
27 SwaraOwa, Coffee and Primate Conservation Project , Java, Central Java , Indonesia
Hoover Kara
Electronic publication date: 2018 Jun 15
Publication date: 2018
Volume: 6
Electronic Location ID: e4869
Received 2018 Mar 27; Accepted 2018 May 10
Copyright: © 2018 Estrada et al.
Copyright year: 2018
Copyright holder: Estrada et al.
License: This is an open access article distributed under the terms of the Creative Commons Attribution License, which permits unrestricted use, distribution, reproduction and adaptation in any medium and for any purpose provided that it is properly attributed. For attribution, the original author(s), title, publication source (PeerJ) and either DOI or URL of the article must be cited.
License URL: https://creativecommons.org/licenses/by/4.0/

Keywords: Deforestation, Logging, Hunting, Poaching, Illegal trade, Protected areas, Agricultural expansion, Community forests, Forest-risk commodity trade, Corruption and governance

Funding: Global Wildlife Conservation CNPq, CAPES and FAPESB (the Research Agency of the state of Bahia) São Paulo State Research Agency (FAPESP Process 2016/08422-0) Brazilian National Science Foundation/CNPq (PQ #303306/2013-0) Institute for Science and Technology (INCT) in Ecology, Evolution and Biodiversity Conservation MCTIC/CNpq (proc. 465610/2014-5) FAPEG ICMBio USAID-CARPE USFWS Anthony B. Rylands and Russell A. Mittermeier received support from Global Wildlife Conservation, Austin, Texas, USA. Sidney Gouveia and Ricardo Dobrovolski are supported by CNPq, CAPES and FAPESB (the Research Agency of the state of Bahia). Their contribution to the manuscript was developed in the context of the Institute for Science and Technology (INCT) in Ecology, Evolution and Biodiversity Conservation, MCTIC/CNpq (proc. 465610/2014-5) and FAPEG. Mauricio Talebi’s contributions were supported by the São Paulo State Research Agency (FAPESP Process 2016/08422-0). Júlio César Bicca-Marques’ contribution was facilitated by the Brazilian National Science Foundation/CNPq (PQ #303306/2013-0). Leandro Jerusalinsky received support from ICMBio. USAID-CARPE and USFWS have contributed to numerous wildlife surveys in DRC and has supported Susan Cheyne’s long-term studies in Indonesia. The funders had no role in study design, data collection and analysis, decision to publish, or preparation of the manuscript.

==============================
Primates occur in 90 countries, but four—Brazil, Madagascar, Indonesia, and the Democratic Republic of the Congo (DRC)—harbor 65% of the world’s primate species (439) and 60% of these primates are Threatened, Endangered, or Critically Endangered (IUCN Red List of Threatened Species 2017-3). Considering their importance for global primate conservation, we examine the anthropogenic pressures each country is facing that place their primate populations at risk. Habitat loss and fragmentation are main threats to primates in Brazil, Madagascar, and Indonesia. However, in DRC hunting for the commercial bushmeat trade is the primary threat. Encroachment on primate habitats driven by local and global market demands for food and non-food commodities hunting, illegal trade, the proliferation of invasive species, and human and domestic-animal borne infectious diseases cause habitat loss, population declines, and extirpation. Modeling agricultural expansion in the 21st century for the four countries under a worst-case-scenario, showed a primate range contraction of 78% for Brazil, 72% for Indonesia, 62% for Madagascar, and 32% for DRC. These pressures unfold in the context of expanding human populations with low levels of development. Weak governance across these four countries may limit effective primate conservation planning. We examine landscape and local approaches to effective primate conservation policies and assess the distribution of protected areas and primates in each country. Primates in Brazil and Madagascar have 38% of their range inside protected areas, 17% in Indonesia and 14% in DRC, suggesting that the great majority of primate populations remain vulnerable. We list the key challenges faced by the four countries to avert primate extinctions now and in the future. In the short term, effective law enforcement to stop illegal hunting and illegal forest destruction is absolutely key. Long-term success can only be achieved by focusing local and global public awareness, and actively engaging with international organizations, multinational businesses and consumer nations to reduce unsustainable demands on the environment. Finally, the four primate range countries need to ensure that integrated, sustainable land-use planning for economic development includes the maintenance of biodiversity and intact, functional natural ecosystems.

Introduction

A recent evaluation of primate species worldwide indicated that more than half are facing near-term extinction due to unsustainable human activities (Estrada et al., 2017). According to the IUCN Red List, wild primates occur in 90 countries across the Neotropics, Africa, and Asia. Sixty-five percent of primate species (286 of 439 species), however, are found in only four countries, —Brazil, Madagascar, Indonesia, and the Democratic Republic of the Congo (DRC) (IUCN, 2017). Based on a comprehensive literature review, we compare the anthropogenic pressures faced by each of these four countries that place primate populations at risk, analyzing differences and similarities affecting land cover changes caused by agricultural expansion, mining and fossil fuel extraction, and local and international trade demands for food and nonfood commodities. We discuss the impact of bushmeat hunting, illegal trade and zoonotic, human and domestic-animal borne infectious diseases on primate population persistence. This information is analyzed within the context of an increasing human population with low levels of human development, income inequality, political instability, and weak governance. We model the expansion of agricultural during the 21st century and identify areas of expected spatial conflict between new crop production and primate distributions in each country. We provide an examination of the conservation value of protected areas, of habitat restoration, and forest connectivity at the landscape level, and stress the importance of community managed forests, where appropriate, for primate conservation at the local level. We further discuss socially oriented conservation actions by NGOs and governments for averting local primate extinction. In our conclusion, we discuss the multiple challenges faced by Brazil, Madagascar, Indonesia, and DRC, as well as the global community to ensure the conservation of their unique primate fauna.

Survey Methodology

We conducted a thorough (at the time of writing) review of the peer-reviewed scientific literature. We integrated the most recent evaluation for primate species conservation status in each country from the International Union for the Conservation of Nature (IUCN, 2017) and information from Global Forest Watch, along with the published literature, to evaluate trends in forest loss between 2001 and 2016 in each country and its effect as a major threat to primate survivorship. Information from FAO (Food and Agriculture Organization of the UN) was used to profile industrial agriculture expansion in the four countries for the same period. We complement these results with a summary of spatial conflict between primate species’ distributions and predicted agricultural expansion during the 21st century for each country. Species distributions were obtained from the IUCN range maps (IUCN, 2017). Agricultural expansion is derived from remote sensing data from IMAGE (Integrated Model to Assess the Global Environment; http://themasites.pbl.nl/models/image/index.php/Agricultural_economy) and represents the predicted presence (irrespective of the intensity) of agricultural production at each grid cell (0.5° of spatial resolution; see Dobrovolski et al., 2014). We document the pressures exerted by international commodities trade on primate habitat loss and degradation using information from the International Trade Centre (http://www.intracen.org/). Legal and illegal primate trade was documented from the CITES (Convention on International Trade in Endangered Species of Wild Fauna and Flora) trade database and from published reports. Information on human population growth and socioeconomic metrics in each country was profiled with information from FAO and the World Bank. Civil conflict and quality of governance indicators for each country were obtained from the 2017 Global Peace Index (GPI) of the Institute for Economics and Peace (http://economicsandpeace.org/) and from the World Bank. We assessed the distribution of protected areas and primate ranges in the four countries using information from the Protected Planet of the UN Environmental Program UNEP-WCMC (2017), the IUCN Red List, and forest cover data from Hansen et al. (2013). We included 2,190 protected areas in the Brazil dataset, 49 in DRC, 147 in Madagascar and 646 in Indonesia (Text S1). We gathered information on the 2016 Corruption Perceptions Index (CPI) of Transparency International (https://www.transparency.org) for each country and obtained, from the World Bank, average values for the four countries of four indicators of governance quality in 2016 (http://info.worldbank.org/governance/wgi/index.aspx#reports). We compared these to the average values for 35 high-income countries.

We are aware that some of the datasets we consulted vary in their level of reliability an objectivity. For example, some data from FAO and the World Bank are based on information provided directly by host governments, and therefore may be incomplete or reflect broad estimates. Similarly, data from the IUCN on the population size, distribution, and conservation status of certain rare, cryptic, or highly inaccessible primate species are based on surveys or census methods that may vary in completeness, and therefore final determinations are subject to a consensus based on “expert opinion.” In other cases, the data are obtained through careful monitoring by an agency (e.g., International Trade Centre, Transparency International) or were independently corroborated using remote sensing to add increased reliability (e.g., Global Forest Watch, IMAGE, Protected Planet). Each of the agencies we used as sources of information stipulate in their portals the limitation of the data they presented (see Text S1 for a list and the relevant URLs). We note that although the numbers reported may vary in their level of accuracy, the trends within and between each country are consistent with high confidence.

Richness of Primate Species and Iucn Threatened Status and Population Status

While Brazil, Madagascar, Indonesia, and DRC differ significantly in their human population demography, culture, history, and economy, they are important reservoirs for the world’s biodiversity, with each considered a megadiverse country (Mittermeier, Robles Gil & Mittermeier, 1997; Table S1; Text S1). They also harbor a nonoverlapping and significant share (65%; n = 286 species) of the world’s nonhuman primate species (n = 439 species): Brazil—102 primate species, 17 genera; Madagascar—100, 15 genera; Indonesia—48, 8 genera; and DRC—36, 15 genera. This includes 55 genera and all 16 recognized nonhuman primate families (IUCN, 2017; Table S2; Fig. 1). Each country’s primate population is imperiled by the expanding pressures of human activities and, as a group, 62% of their primate species are Threatened (i.e., assessed as either Vulnerable, Endangered or Critically Endangered on the IUCN Red List) and 72% are declining (IUCN, 2017; Fig. 1). The two countries with the greatest number of Threatened and declining primate species are Madagascar and Indonesia followed by Brazil and DRC (Fig. 1).

Figure 1 The richness of species and IUCN species conservation and population status of primates in Brazil, the Democratic Republic of the Congo (DRC), Madagascar and Indonesia.

In the graph, the numbers below the names of the countries refer to the number of species used to calculate the percentages for species threatened and declining populations. Because population assessments are not available for all species, we focused on those for which recent information is available (Table S2). Source of data: IUCN Red List 2017-3 (http://www.iucnredlist.org; accessed 5 February 2018).

Large-scale encroachment and loss of primate habitats

Trends in forest loss

Habitat loss is a major driver of local extirpation of primate species. Using information from the Global Forest Watch database (GFW, 2018; Hansen et al., 2013) we found a general increase in loss of forest (defined as >30% canopy cover), for the period 2001–2016 in all four countries (Fig. 2A). Total forest loss for the period was 46.43 M ha for Brazil, 23.08 M ha for Indonesia, 10.52 M ha for DRC, and 2.75 M ha for Madagascar (Fig. 2B; Table S3). Brazil’s initiatives to combat deforestation resulted in important reductions in forest loss (80%) from 2005 to 2012 (Fig. 2; Nepstad et al., 2014; PRODES, 2018), although in biomes such as the dry forests of the Cerrado, deforestation continued at high rates (Strassburg et al., 2017). Unfortunately, deforestation in Brazil increased sharply in 2016 (Figs. 2A and 2B), probably the result of a shift in government policies that have relaxed conservation laws (Brancalion et al., 2016).

Figure 2 (A) Trends in tree cover loss (>30% canopy cover) in Brazil, DRC, Indonesia, and Madagascar for the period 2001–2016. (B) Cumulative tree cover (in Intact Forest Landscapes IFL) loss in each country for the same period. Source of data Global Forest Watch (http://www.globalforestwatch.org; accessed 5 February 2018). IFL: an unbroken expanse of natural ecosystems of at least 500 km2, forested, and without signs of significant human activity (Potapov et al., 2008). Forest loss ranged in Brazil from 2.74 M ha in 2001 to 5.37 M ha in 2016; in Indonesia from 745.43 K ha to 2.42 M ha; in DRC from 455.43 K ha to 1.38 M ha, and in Madagascar from 86.95 K ha to 383.55 K ha.

Importantly, between 2000 and 2013 each of the four countries experienced losses in their remaining area of Intact Forest Landscapes (IFL; Potapov et al., 2017). The largest percent of IFL losses occurred in Madagascar and Indonesia, followed by Brazil and DRC (Table 1). These trends highlight important reductions in primate habitats that are exacerbated by increases in low-density, small-scale deforestation, which is more difficult to identify and track (Kalamandeen et al., 2018). For example, in Amazonia, the number of new small clearings (<1 ha) increased by 34% between 2001 and 2007 and small-scale low-density forest loss (km2 forest loss per 100 km2) expanded markedly between 2008 and 2014. Overall, cleared forest patches less than 6.25 ha accounted for ∼34% of the total Brazilian Amazon forest lost between 2001 and 2014, including forest loss in reserves that are described as protected areas (Kalamandeen et al., 2018). In 2000, DRC was reported to have almost 2 M km2 of forest (>30% canopy cover) (GFW, 2018). Of this, 32% was classified as Intact Forest Landscape (Potapov et al., 2017) and (36%) as hinterland forests (minimally disturbed forests, Tyukavina et al., 2013). Between 2000 and 2013, 4.2% of DRC’s intact forests were lost (Table 1; Potapov et al., 2017), and in total 5.3% of the country’s total forest was lost between 2001 and 2016 (GFW, 2018). Over the past five years, DRC has experienced a mean annual forest loss of approximately 0.5%, the lowest of the four countries in this analysis.

Table 1 Tree cover loss (30% canopy cover) in Intact Forested Landscapes in Brazil, DRC, Indonesia, and Madagascar for the period 2001–2016.

	Forest cover (>30% canopy, 2000; km2 × 103)	IFL area 2000 (km2 × 103)	% of IFL of country’s forest cover in 2000	Reduction 2000–2013 (%) not attributed to fire	
Madagascar	170	17.2	10	18.5	
Indonesia	1,610	359.2	22	10.8	
Brazil	5,190	2476.1	48	6.2	
DRC	1,992	643.9	32	4.2	
Note:

Source of data: Potapov et al. (2017).

Wide range tropical deforestation also results in forest fragmentation, leading to higher extinction rates in local populations (Hanski et al., 2013). A recent study predicts that additional forest loss will result in a large increase in the total number of forest fragments in the Neotropics, Africa and Asia, accompanied by a decrease in their size (Taubert et al., 2018). In general, extinction risk increases with decreasing fragment size (Hanski et al., 2013).

Trends in expansion of agricultural land

Keeping in mind the limitations of statistics reported by the FAO of the United Nations (information provided to the FAO comes directly from host governments who may provide incomplete data), from 2001 to 2015 the combined estimated increase of agricultural land in Brazil, Madagascar, Indonesia, and DRC totaled some 29.5 M ha (see Text S1; Fig. S1; Table S4), with Brazil having the largest increase (19.1 M ha) followed by Indonesia (9.3 M ha), DRC (650 K ha), and Madagascar (572 K ha) (Table S4) (for estimates of trends in the production of key crops in each country for the period 2001–2015 see Figs. S2–S5 and Text S1). The agricultural footprint (increase of agricultural area as percent of land area, based on data from FAO and the World Bank; Table S4) for this period was 4.89% for Indonesia, 2.25% for Brazil, 0.97% for Madagascar, and 0.28% for DRC. In the case of DRC, a higher footprint estimate of 1.20% has been reported for rural areas (period from 2000 to 2010) resulting in the addition of 2.77 M ha of rural roads, villages, and active and abandoned fields and gardens. This rural complex accounted for 13.1% of DRC’s total land area in 2015 (Molinario, Hansen & Potapov, 2015). Between 2000 and 2010, the overall loss of “core forest” (which made up 36.6% of the 2010 land area) to perforated forest, patch forest, fragmented forest or edge was estimated at 3.8% (Molinario et al., 2017). The main cause of forest loss in DRC (92%) was shifting cultivation (Molinario et al., 2017).

Projected agricultural expansion and primate range contraction in the 21st century

Increases in species extinction risk are typically related to the loss of individual populations and associated declines in their geographical range (Ceballos & Ehrlich, 2002; Wolf & Ripple, 2017). A global study modeling conflict between agricultural expansion and primate species’ distributions predicted that during the 21st century, regions expected to be converted from forest to agricultural production account for 68% of the area currently used by primates, and that worldwide this will lead to unsustainable spatial conflict for 75% of primate species (Estrada et al., 2017). Modeling agricultural expansion in the 21st century for the four countries under a worst-case-scenario, shows a primate range contraction of 78% for Brazil, 72% for Indonesia, 62% for Madagascar, and 32% for DRC (Figs. 3 and 4). A business-as-usual scenario also predicts high spatial conflict while an optimistic scenario predicts significantly lower spatial conflict (Fig. S6). This suggests that targeted policies designed to shift agricultural expansion to already altered landscapes in order to minimize habitat fragmentation and loss of existing forest is critical in limiting spatial conflicts in each country (Dobrovolski et al., 2013, 2014). Global dietary changes, towards eating more meat, greater dependence on vegetable oils, and, to a lesser extent, more coffee and tea, as countries develop, will require these primate-rich countries to convert additional forested land into monocultures to meet local and global market demands (Gouel & Houssein, 2017; Kastner et al., 2012; Tilman & Clark, 2014). Other threats such as hunting, logging, mining, fossil fuel extraction, anthropogenic infectious diseases, and climate change also are expected to result in primate range contraction (see below).

Figure 3 The projected expansion of agriculture and pastures in (A) Brazil, (B) the Democratic Republic of the Congo, (C) Madagascar, and (D) Indonesia for 2050 and 2100, under a worst-case scenario of land use from native vegetation to agricultural fields and pasture.

See Text S1 for a description of the methods used. Data on species geographic distribution are derived from IUCN (2017) and the scenarios of agricultural expansion from the Integrated Model to Assess the Global Environment (IMAGE, version 2.2) (IMAGE Team, 2001) (see Dobrovolski et al., 2013). Notice the spatial shift of conservation conflicts, including the abandonment of some agricultural areas by 2100 in DRC and Madagascar. This condition, however, may not imply an immediate benefit for primate species, as local populations would have been extirpated, areas would have been dramatically altered prior to abandonment, and would likely require decades to regenerate to closed-canopy, old secondary forest. See Fig. S6 for a model based on an optimistic scenario and on a business-as-usual scenario.

Figure 4 Photos of selected land cover changes in primate range countries, illegal primate trade, and the primate bushmeat trade.

Photo credits include the following: (A) Soybean plantation and recent deforestation of forest patches in the Cerrado Biome, Jataí, Goiás State, Brazil (Photo credit: Fabiano R. de Melo), (B) Pastures for cattle ranching surrounding Atlantic Forest patches inside the Cerrado Biome, Rio Verde, Goiás State, Brazil. (Photo credit: Izaltino Guimarães Jr), (C) Indonesia, illegal logging Central Kalimantan (Photo credit: R. Butler), (D) Indonesia, deforestation (Photo credit: R. Butler), (E) Indonesia, Sunda slow loris (Nycticebus coucang), sold in Jakarta (Photo credit: A. Walmsley and Little Fireface Project), (F) DRC, smoked bonobo (Pan paniscus) meat at a rural meat market (Photo credit: J. Head).

Other Large-Scale Stressors

Logging, mining and fossil fuel extraction and primate habitat loss and degradation

Since the 1980s, the extraction of hardwoods has increased in the four countries in response to an ever-expanding worldwide demand for tropical timber (Estrada, 2013). This has resulted in deforestation and new economic incentives to construct roads in forested areas (Alamgir et al., 2017). Although some primate species can survive temporarily in logged forests, both legal and illegal logging result in a decrease of canopy cover, reduced humidity in the subcanopy and undergrowth that increases tree mortality, the incidence of ground fires, a decline in forest undergrowth, and negatively impacts the regeneration of large tree species that provide food, resting sites, and refuge for primates (Alisjahbana & Busch (2017); Lewis, Edwards & Galbraith, 2015; Peres, 1999, 2001; World Bank, 2016) (Text S1).

Mining is a persistent threat to primates and their habitats. The mining of precious gems and minerals contributes to habitat destruction, fragmentation, deforestation, and the poisoning and pollution of soil and ground water (Alvarez-Berríos & Aide, 2015). In addition, mining (and fossil fuel extraction, see below) stimulates human migration, the illegal logging and colonization of forested areas, hunting, and the construction of roads and railways (Alamgir et al., 2017; Butt et al., 2013; Laurance et al., 2015). In eastern DRC, there is an unfortunate overlap of unprotected areas of high animal and plant biodiversity with areas that are rich in minerals (Edwards et al., 2014). Increased global demand for easily-mined surface deposits of tantalum, a rare earth metal used in electronics including cell phones, has resulted in the expansion of illegal mining camps in several national parks in DRC. Bushmeat hunting in this area has decimated several primate populations (e.g., Grauer’s gorillas, and eastern chimpanzees; Plumptre et al., 2015; Spira et al., 2017). Of the existing 1,249 mining prospection permits in DRC, 952 (76%) have their centers in the rural complex (areas that have been in the cycle of slash-and-burn agriculture for at least 18 years). Permits in the rural complex cover 143,316 km2, which is 78% of the total permitted area. The mean area of mining permits is 150 km2 (and there is no difference between the size of permitted area in the rural complex and in forests more distant from human settlement). Approximately one-quarter of the mining prospection permits are located inside the forest and, if these are opened up for mineral extraction, they will pose a grave threat to primates (see Text S1).

In Madagascar, the illegal mining of nickel, cobalt, gold, and precious gems (sapphire) has affected many forests, including protected areas with an important negative impact on populations of Malagasy primates including the iconic ring-tailed lemur (Lemur catta) (Gould & Sauther, 2016). In Brazil, between 2001 and 2013 approximately 1,680 km2 of tropical moist forest was lost across 1,600 gold mining sites, including significant forest loss inside 13 protected areas (Alvarez-Berríos & Aide, 2015). A more recent study showed that between 2005 and 2015 mining in Brazil significantly increased Amazon forest loss up to 70 km beyond mining lease boundaries, causing 11,670 km2 of deforestation (9% of all Amazon forest loss during this period) (Sonter et al., 2017). The disposal of mining waste is a significant threat to the local biota, including primates. In Brazil, for example, 126 mining dams are currently at risk of failing. In one such case, dam failure poisoned hundreds of kilometers of the Doce River with toxic mud (Garcia et al., 2017). In Kalimantan, Indonesia, gold mining is a major threat to the proboscis monkey (Nasalis larvatus) (Meijaard & Nijman, 2000) and to Bornean orangutans and Bornean gibbons (Hylobates muelleri) (Lanjouw, 2014). From 2000 to 2010, some 3,000 km2 of, mostly lowland, forest in Indonesia was lost due to logging and as of 2011, over 40,000 km2 of additional land was allocated to mining concessions (Abood et al., 2015). Most of these concessions are located on the islands of Sumatra and Borneo, where it directly impedes with conservation efforts to protect arboreal primates such as the slow loris (Nycticebus spp.), langurs (Presbytis spp. and Trachypithecus spp.), gibbons (Hylobates spp.), siamangs (Symphalangus syndactylus) and orangutans (Pongo spp.). For some species such as the western tarsier (Tarsius bancanus) and Sody’s slow loris (Nycticebus bancanus) on the island of Belitung (Yustian, 2007), finding a way to manage tin mines using environmentally friendly approaches is crucial for the survival of these nocturnal primates. In addition, traditional methods of gold mining and limestone karst mining now threaten the habitat of the agile gibbon (Hylobates agilis), the siamang, the black-crested Sumatran langur (Presbytis melalophos) and the silvered langur (Trachypithecus cristatus) in the province of Jambi in West Sumatra. Miners living in these areas also exploit primates and other wildlife for meat and capture live primates for pets that are sold in local towns (see hunting and illegal trade below; Agustin et al., 2016; Yanuar, 2009).

Fossil fuel extraction negatively impacts primate survivorship. For example, over the next 20 years, the global demand for oil is expected to increase by over 30% and the expected increase in natural gas by 53% from 2014 levels (Butt et al., 2013; Finer et al., 2015). This peak oil production it is projected to fall to present day levels (due to the changeover to electric vehicles) by the year 2040 (Longley, 2018). Brazil, Indonesia, and Madagascar are already expanding concessions and exporting this commodity (The International Trade Center–www.intracen.org). In the western Amazon of Brazil, for example, such concessions include national parks and territories of indigenous peoples (Finer et al., 2015). In DRC, oil concessions now cover almost all of the Albertine Rift and much of the central basin, where a concentration of endemic primate taxa is found (Ministère des Hydrocarbures DRC, 2013).

International commodities trade and loss and degradation of primate habitat

International trade commodity-driven deforestation is increasingly caused by global demand for agricultural and nonfood commodities (e.g., soy, beef, palm oil, timber, ores, fossil fuel) negatively impacting tropical biodiversity (Henders, Persson & Kastner, 2015; Henders et al., 2018; Wich et al., 2014) and primate range and population persistence (Estrada et al., 2017). While the growing human populations in Brazil, Madagascar, Indonesia, and DRC (see Human Population below) have resulted in increased internal demands for food and non-food commodities, global market pressures from highly industrialized nations are significant drivers of rapid and widespread habitat loss. According to the International Trade Centre, these four primate-rich countries sell at least 50% of all exports of raw materials to China, the US, Canada, India, and several European countries (Table 2). Commodities such as frozen beef, soy, sugar cane, hardwoods, and ores are principal exports of Brazil; in DRC minerals are the primary global export commodity, followed by smaller amounts of hardwoods, natural rubber, coffee, and cacao; for Madagascar, major exports are minerals, coffee, tea, spices, hardwood, and vegetable and roots/tubers; and for Indonesia, rice, natural rubber, oilseeds, and wood (Text S1). In Brazil, 30% of deforestation between 2000 and 2010 was driven by global demands for beef and soy exports (Karstensen, Peters & Andrew, 2013). Given that segments of the human population in each of these countries are undernourished (see Human Population below), the exportation of food may threaten local food security, human safety and political stability (FAO, IFAD & WFP, 2015). The growing and unsustainable global demand for food and non-food crops, wood, fossil fuel, minerals, and gems by a small number of consumer nations has resulted in a rapid increase in agricultural production, wood extraction, itinerant miners, and oil/gas extraction. This also has led to an expansion of road networks and hydropower development in all four countries (Alamgir et al., 2017), ensuing increased forest loss, illegal colonization and logging, increases in itinerant mining and increases in primate hunting and trade (Estrada et al., 2017; Latrubesse et al., 2017; Plumptre et al., 2015; Spira et al., 2017; Timpe & Kaplan, 2017; Winemiller et al., 2016). Importing nations process the raw materials and the final product is commercialized for local and global consumption. A particulary unfortunate example of this is the growing global demand for products produced by industrialized nations such as cell phones, laptops, and other electronic devices using conflict minerals such as coltan, mined in DRC (Hayes & Burge, 2003; Mancheri et al., 2018; Spira et al., 2017). To balance global market demands with the needs of the four primate-rich countries to develop their internal economies, ensure food security, and improve the standard of living for their expanding human populations, the “greening” of trade can promote environmental protection (Neumayer, 2001; Henders et al., 2018). International corporations should add these costs to products so that there is a continuous regeneration of funds to sustainably promote conservation (Butler & Laurance, 2008). Alternatively, the World Bank or UN could require that corporations and consumer nations pay into a sustainability/conservation fund based on their levels of consumption and environmental damage (e.g., like a carbon tax; Carbon Tax Center https://www.carbontax.org; consulted August 2017). In countries in which the rural poor depend on forest products, community forest management could bridge or integrate the needs of conservation and commodity production, sustainably safeguarding the continued integrity of complex ecological systems (Sharif & Saha, 2017). The recent environmentally-oriented, demand-side policies regarding illegal timber imports by the EU (EU, 2010), the EU resolution on oil palm production and deforestation (EP, 2017), and the Amsterdam Declaration to eliminate deforestation from agricultural commodity chains (Amsterdam Declaration, 2015) represent important and positive “green” changes that need to be adopted by the U.S., China, and other consume nations. However, the continued growth of the global demand for forest-risk agricultural and nonfood commodities requires additional legislation and a stronger global effort at regulating the negative impact of unsustainable commodity trade (Henders et al., 2018).

Table 2 Major importing countries (50% of exports) of trade commodities (99 categories and their subcategories, e.g., frozen beef, arboreal and non-arboreal food and non-food crops, ores, oil, wood, and others) produced by Brazil, DRC, Madagascar, and Indonesia.

	Brazil		DRC		Madagascar		Indonesia	
	%Volume imported by		%Volume imported by		%Volume imported by		%Volume imported by	
China	19	China	46	France	24	China	19	
USA	13	S. Arabia	11	USA	13	USA	11	
Argentina	7			Germany	9	Japan	11	
The Netherlands	6			China	7	India	8	
Germany	3							
Japan	3							
Total %	50		57		53		50	
Note:

Source of data: (http://www.trademap.org/ (accessed 10 December 2017)). International trade maps for the four countries for all exports and for specific commodities see Text S1.

Local-Scale Anthropogenic Threats to Primate Populations

Hunting

Hunting (for meat and culturally valued body parts) negatively impacts 54% to 90% of primate species in the Neotropics, Africa, Madagascar, and Asia (Estrada et al., 2017). According to IUCN, about 85% of primate species in Indonesia are hunted, 64% in Madagascar, 51% in DRC, and 35% in Brazil (IUCN, 2017), but we need to recognise that the IUCN primate assessments are now 10 years old and many do not mention hunting specifically. The new assessments of the African primates (which will come online in 2018) are in general much clearer regarding individual threats and a much higher percentage—at least in Africa—will list hunting as a primary threat than in previous assessments. In reality, for example, almost all primates in DRC are hunted—even the smallest monkey, the talapoin has now been recorded at bushmeat markets (Bersacola et al., 2014). An exception is the nocturnal strepsirrhines, which are so small and so hard to catch that they are rarely taken unless for traditional medicine. Commercialized bushmeat hunting is a primary driver of primate population reduction and, in the case of the Brazilian Amazon, has led to the extirpation of highly endangered taxa such as spider monkeys (Ateles spp.) and woolly monkeys (Lagothrix spp.) (Effiom et al., 2013; Peres et al., 2016; Stevenson & Aldana, 2008). Hunting has contributed to extirpation of smaller and threatened primates in Brazil’s Atlantic Forest such as the yellow-breasted capuchin monkey (Sapajus xanthosternos), Coimbra-Filho’s titi monkey (Callicebus coimbrai) (Canale et al., 2012; Hilário et al., 2017) and the largest Neotropical primate species, the southern muriquis (Brachyteles arachnoides) (Talebi et al., 2011). In DRC hunting has significantly reduced the numbers of gorillas and bonobos (Hickey et al., 2013; Plumptre et al., 2016c). In a wild meat market in Kisangani (DRC) about 65 primates were traded per day over a 131-day period (about 8, 515 primates/131 days) (Van Vliet et al., 2012). In Basankusu (DRC), the rate was 17 primates traded per visit (Dupain et al., 2012). The primates present in these markets included species of the genera Chlorocebus, Cercocebus, Colobus, as well as chimpanzees and bonobos (Text S1). In DRC, the Endangered or Critically Endangered l’Hoest’s Monkey (Allochrocebus lhoesti), Dryas monkey (Cercopithecus dryas) (Fa et al., 2014), Grauer’s gorilla (Gorilla beringei graueri) and the eastern chimpanzee (Pan troglodytes schweinfurthii) experience high levels of poaching and are part of the commercial bushmeat trade (Fig. 4; Plumptre et al., 2015, 2016a, 2016b, 2016c; Spira et al., 2017). In DRC, hunting has resulted in emptying of all but the smallest bodied faunal species across large swathes of forest. For example, a large area of the Sankuru Natural Reserve has almost no bonobos remaining (Liengola et al., 2009); in a survey of the corridor area between the two sectors of the largest national park in the country (Salonga), bonobos were never found closer than 10 kilometers from the nearest village (Maisels, Nkumu & Bonyenge, 2009; see Text S1). Given that only 21–27.5% of bonobos live in protected areas (Hickey et al., 2013), their survival into the next century remains in doubt. However, primates living in protected areas also face significant challenges. Most of the remaining 3,800 Grauer’s gorillas and all mountain gorillas (Gorilla beringei beringei estimated population size 880) are restricted to protected areas (Plumptre et al., 2016b). Because the population density of lemurs, monkeys, and apes living outside of protected areas has decreased rapidly, this has resulted in an increase in the price or value of primate bushmeat, making it profitable for hunters to risk prosecution by entering into protected areas (Rovero et al., 2012).

Poorer households in the forested northwestern Makira landscape of Madagascar rely more on wildlife than richer households (Golden et al., 2016). Widespread hunting of black-and-white ruffed lemurs (Varecia varecia), diademed sifakas (P. diadema) and the brown lemur (Eulemur fulvus) in eastern Madagascar, has put these primates at increased risk (Jenkins et al., 2011). In periods following political crisis and instability in Madagascar, lemurs were traded as a prized source of meat (Barrett & Ratsimbazafy, 2009). Larger diurnal species such as the black-and-white ruffed lemur, indri (Indri indri), and sifaka (Propithecus spp.) are targeted because traditional taboos protecting lemurs have eroded rapidly (Golden, 2009; Jenkins et al., 2011). Even small species such as mouse lemurs (Microcebus spp.) are eaten, with hunters capable of capturing up to 50 a night; the impact on wild populations is considerable (Gardner & Davies, 2014) (Text S1). Primate bushmeat consumption and trade in southern Sumatra results in hundreds of macaques killed monthly to meet the demand from wild meat restaurants (KSBK, 2002). Other primates eaten are the Sangihe Island tarsiers (Tarsius sangirensis; Shekelle & Salim, 2009) and Bornean orangutans (Pongo pygmaeus; Meijaard et al., 2011). In Borneo, between 1,950 and 3,100 orangutans are killed annually for consumption (including 375–1550 females), significantly impacting the viability of many small isolated populations (Ancrenaz et al., 2015; Ancrenaz et al., 2016; Meijaard et al., 2011; Santika et al., 2017a). In Indonesia, even subsistence hunting can have major effects on primate populations already decimated by land conversion and habitat loss (orangutans in Sumatra, Kloss’ gibbons, pig-tailed langurs, Mentawai Island langurs and populations of Trachypithecus spp. and Presbytis spp. on others Indonesian islands) (Fuentes, 1998, 2002; Paciulli, 2004).

Numerous primates in each of the four countries consume ripe fruits and serve as important agents of seed, dispersal promoting forest regeneration (Chapman et al., 2013). The extirpation of primates due to hunting results in a change in dispersal dynamics, the size and distribution of seed shadows, a reduction in plant genetic diversity and seedling recruitment (Caughlin et al., 2015; Pacheco & Simonetti, 2000; Brodie et al., 2009). There also is evidence that lemur population decline has resulted in the reduced viability of several species of Malagasy trees (Federman et al., 2016). Similarly, the population collapse of larger-bodied primates in response to over-hunting in the Brazilian Amazon has impacted the regeneration of long-lived and hardwood tree species and this is likely to reduce the ability of these forests to store carbon (Peres et al., 2016; Stevenson & Aldana, 2008). The overhunting of primates reduces the recruitment of trees whose seeds they disperse which also reduces food sources available to the local mammalian and avian communities (Abernethy et al., 2013; Nuñez-Iturri, Olsson & Howe, 2008).

Legal and illegal primate live trade

Many primate species are impacted by unsustainable live trade, often organized by criminal networks or sanctioned by local and national governments (Fig. 4, Alves, Souto & Barboza, 2010; Nijman et al., 2011; Alamgir et al., 2017; Nekaris et al., 2013; Shanee, Mendoza & Shanee, 2015; UNODC, 2013). According to the CITES trade database, Indonesia is the leading exporter of live primates, with 98% being either captive-bred or captive-born long-tailed macaques (Macaca fascicularis) and the remainder principally wild-caught animals from a number of other species (Table 3). Most of the international trade from Indonesia is for scientific or biomedical research (V. Nijman, 2017, unpublished data based on CITES trade data). In DRC, over the last decade a much smaller number (N = 581) of primates, mostly guenons (Cercopithecus spp.), were exported for purposes of commercial trade, and almost all were wild-caught. (V. Nijman, unpublished data based on CITES trade data). However, there appear to be wide discrepancies between the numbers reported by the importing countries (N = 561) and the numbers reported as exported by DRC (N = 347) (other items, such as skin, bones, “specimens” total 16,202 reported by importers; DRC reported 5,364 exports over the same period). In contrast, the live primate trade out of Madagascar and Brazil appears to be better controlled, with only 24–51 individuals, bodies and skins reported. All primates exported from Madagascar were wild-caught (Table 3).

Table 3 CITES trade from Indonesia, Brazil, DRC, and Madagascar over the period 2006–2016 (data from 2016 incomplete).

Country	Indonesia	Brazil	DRC	Madagascar	
Live animals					
 Importer	15,579 (0.06)	166 (0)	561 (100)	13 (7.69)	
 Exporter	19,009 (0.67)	154 (0)	217 (97.24)	4 (25.00)	
Bodies, skeletons, skins					
 Importer	40 (100)	0 (0)	20 (90.00)	11 (100)	
 Exporter	3 (0)	153 (60.13)	9 (100)	47 (100)	
Specimens					
 Importer	51,743 (12.65)	385 (82.60)	4,876 (92.99)	17,695 (100)	
 Exporter	73,780 (33.06)	2,449 (60.76)	4,184 (93.40)	10,805 (99.96)	
Note:

Percentage of wild-caught in brackets. Importer refers to data as reported by the various importing countries; exporter refers to data reported by the exporting countries, here Indonesia, Brazil, DRC, and Madagascar. Source: https://trade.cites.org/ (accessed 15 August 2017). See Text S1.

In general, the illegal trade in primates is for pets, meat, and medicinal or mystical purposes. In Brazil, legal international trade in live primates appears to be limited (Svensson et al., 2016). However, surveys of animal markets in Brazil and in the tri-country border of Peru–Colombia–Brazil showed that capuchin and brown woolly monkey (Lagothrix lagothricha) body parts were important trade items (Ferreira et al., 2013; Van Vliet et al., 2014) (Text S1). The pet trade in primates in Indonesia occurs openly in dozens of markets, and is prevalent in Sumatra, Java, and Bali, as well as in Indonesian Borneo and on Sulawesi. For example, during 66 visits to bird markets in North Sumatra, 10 species of primates totaling 1,953 individuals were available for sale (Shepherd, 2010). Some 1,300 primates were recorded during 51 surveys to six markets on Java and Bali (Nijman et al., 2017). This included individuals of eight species. The most common primates traded were macaques and the greater slow loris (Nycticebus coucang, Text S1). Slow lorises are locally traded for medicinal purposes throughout Indonesia (Nekaris et al., 2010) (Text S1). In Madagascar, a study reported the presence of ∼30,000 pet lemurs of at least 16 species over a three-year period (Reuter et al., 2016).

Harvesting (capture and killing) to extinction

Range contraction, combined with unsustainable bushmeat hunting and capture for the trade of selected species, suggests that high prices for rare or difficult to acquire species can, over time, drive even large populations to local extirpation. The Anthropogenic Allee Effect (Courchamp et al., 2006) proposes that such extinctions are caused when prices for wildlife products increase with species rarity and that this price-rarity relationship creates financial incentives to extract the last remaining individuals of a population, despite higher search and harvest costs (Holden & McDonald-Madden, 2017). Another study suggests that while range contraction (habitat loss and fragmentation) causes population declines, local densities may remain relatively stable, especially in the case of animals like primates in which individuals can live for 20, 30, or >40 years, facilitating harvesting to extinction of selected species (Burgess et al., 2017). The authors also showed that opportunistic exploitation, where harvesters hunt or capture rare species while chasing target species, can significantly reduce population number. Clearly, current and predicted range contraction and abundance declines increase the extinction risk to harvested primate species in the four countries. This deserves greater consideration in research, conservation management, and protection plans.

Other Emerging Threats

Infectious diseases

Across anthropogenically impacted landscapes, the threat to primates of exposure to emerging infectious diseases resulting from increased contact with human and domesticated animals or periodic epizootic outbreaks across a broad region can result in local primate population declines or extirpations from otherwise suitable habitat (Hoffmann et al., 2017; Nunn & Altizer, 2006; Nunn & Gillespie, 2016). Between October 2002 and January 2004, outbreaks of EVD (Ebola Virus Disease) killed over 90% of the western gorillas (Gorilla gorilla) and possibly 80% of chimpanzees inhabiting the Lossi Sanctuary in northwest Republic of Congo (Bermejo et al., 2006). To date, however, there has not been an Ebola outbreak associated with any species of wildlife in DRC (Pigott et al., 2014, 2016). Developing vaccines that can be administered safely and effectively to free-ranging populations of great apes may help mitigate the impact of EVD outbreaks although this would be extremely challenging since these primates are hunted and hence are not habituated to humans (Leendertz et al., 2017). In most cases, these vaccines are not yet available even to local human populations, which presents an ethical dilemma regarding whether or not to provide these vaccines to endangered apes. In Brazil, 80% of isolated populations of black-and-gold (Alouatta caraya) and brown (Alouatta guariba clamitans) howler monkeys in two areas in the state of Rio Grande do Sul were lost after a Yellow Fever (YF) epizootic event in 2008 and 2009 (Almeida et al., 2011; Freitas & Bicca-Marques, 2011; Vasconcelos, 2017; Veiga, Fortes & Bicca-Marques, 2014), including populations inhabiting protected areas (Fialho et al., 2012). Since 2016, an ongoing YF outbreak in Southeast Brazil has caused the death of thousands of primates, including threatened species such as the northern masked titi monkey (Callicebus personatus) and the brown howler monkey. In many instances, misinformation regarding vectors of YF disease transmission has resulted in members of the local human population exterminating nearby monkey populations (Bicca-Marques et al., 2017) (Text S1).

Susceptible primate populations inhabiting protected areas also are vulnerable to the introduction of exotic (non-native or alien) pathogenic agents into the naïve population, a process known as pathogen pollution (Daszak, Cunningham & Hyatt, 2000). The death of introduced marmosets (Callithrix spp.) infected with human herpesvirus 1 in a Brazilian nature reserve illustrates how proximity to humans can risk the survival of wild primate populations (Longa et al., 2011). The risk of epizootic disease transmission is particularly serious for those primates living near or within regions inhabited by dense human populations, such as in most of Indonesia, where Streptococcus equi caused high mortality among long-tailed macaques in 1994 (Soedarmanto et al., 1996). In Indonesia, outbreaks of measles, rubella, and parainfluenza have affected the survivorship of long-tailed macaque (M. fascicularis) groups living in close contact with humans (Schillaci et al., 2006). In Madagascar, lemurs inhabiting forests near human settlements are exposed to pathogenic enterobacteria (E. coli, Shigella spp., Salmonella enterica, Vibrio cholera and Yersinia spp.; Bublitz et al., 2015), protists (Cryptosporidium sp.; Rasambainarivo et al., 2013; Toxoplasmosis gondii) and viruses (Herpesvirus hominis and West Nile Flavivirus; Junge & Sauther, 2006) found in humans, livestock, pets and peridomestic rodents. Likely or proven cases of transmission of human diseases to great apes include enterobacteria, human herpes simplex virus, a measles-like disease, a polio-like disease, respiratory diseases, and scabies (Gilardi et al., 2015).

Climate change

Evidence for the impact of local and global climate change on primate populations is limited. However, current assessments indicate the expected extremes in temperature and rainfall will put primates at significant risk (see Fig. 2 of Graham, Matthews & Turner, 2016). Climate change projections suggest that Brazil’s four endemic species of Atlantic forest lion tamarins (Leontopithecus spp.) will experience major shifts and/or reductions in habitat suitability in the coming decades (Meyer, Pie & Passos, 2014). Similarly, the distribution of the northern muriqui (Brachyteles hypoxanthus) is expected to be reduced by more than half of its present area, with a large decline in the future suitability of currently protected reserves due to climate change (Melo et al., 2016). In Madagascar, in response to climate change most lemur species are expected to experience marked reductions in population number and distributions, even in the absence of future anthropogenic deforestation, with predicted declines of ∼60% for lemurs’ habitats (Brown & Yoder, 2015).

Climate change will likely increase primate exposure to potentially harmful human-borne parasites, triggered, for example, by increases in temperature and rainfall leading to faster parasite reproduction or longer periods of parasite transmissibility in primate rich regions (Barrett et al., 2013). Although certain species may be successful in shifting their range into newly created or expanded environments, this is likely to have negative consequences for other species that are displaced or out competed (Schloss, Nuñez & Lawler, 2012). For example, forest fragmentation resulting from changing climates is expected to limit the availability of dispersal routes used by titis (Callicebus spp.) in eastern Brazil (Gouveia et al., 2016). Moreover in the future, protected areas and parks created to sustain threatened species may no longer be suitable due to changes in vegetative cover in response to climate change, or individuals may migrate into neighboring and unprotected forests where they are exposed to hunters or local residents (Araújo et al., 2004; Malhi et al., 2008; Struebig et al., 2015; Wiederholt & Post, 2010). Projections of climate change in Central Africa are less clear (Abernethy, Maisels & White, 2016). However, rainfall decline may occur, leading to a reduction in forest cover in DRC (Beyene, Ludwig & Franssen, 2013); other work suggests the opposite may be true (Zelazowski et al., 2011). Regardless, clearing of additional forest for agriculture results in land desiccation which when combined with droughts and El Niño episodes result in extensive wildfires (Laurance, Sayer & Cassman, 2014; World Bank, 2017), impacting primate populations (Graham, Matthews & Turner, 2016). The most forceful example of this is human-made fires that resulted in the burning of 2.6 M ha of land in Indonesia between June and October of 2015. These fires were fed by drought and the effects of a prolonged El Niño. Degraded peatlands, most of them found in Sumatra, Kalimantan, and Papua Province, Indonesia are particularly sensitive to fires that easily spread to adjacent forests. For example, the 2015 fires burned some 700 K ha of natural forest, swamp forest and forestry concessions plus 505.8 K ha of palm oil concessions (World Bank, 2017). Therefore, mitigating climate change impacts on the potential for mass fires is critical for primate survivorship in Indonesia.

Human Population

Trends and projections in human population growth

Environmental pressures exerted by a growing human population are a major driver of primate habitat and population decline in each country (Crist, Mora & Engelman, 2017). In 2016, Indonesia was the most populous of the four countries with slightly over 263 million people, followed by Brazil (about 211 million), DRC (about 80 million), and Madagascar (about 26 million). Human population density is highest in Indonesia (145 people/km2) and lowest in Brazil (25 people/km2) (Table 4; Text S1). Population growth rates for 2016 were highest in DRC (3.09%/yr) and Madagascar (2.75%/yr), lower in Indonesia (1.07%/yr) and lowest in Brazil (0.77%/yr) (Table 4). Human population projections for the year 2050 indicate continued growth in all four countries with DRC showing the steepest increase, followed by Madagascar, Indonesia, and Brazil. In Brazil and Indonesia, much of this population growth is expected to occur in urban areas (Fig. 5). Also, although in the short term rural populations are expected to expand rapidly in DRC and Madagascar, projections suggest that by 2050 their urban population (69% of the population of DRC and 55% of the population in Madagascar) will surpass their rural population (Fig. 5). The large size and projected increase of the population in all four countries in the first half of this century is expected to exponentially extend the human and urban footprint on primate habitats, near and beyond cities. These negative impacts will result from increasing demands for energy, space, food, water, minerals, oil, construction material, forest products, and transportation, as well as from environmental damage caused by pollution and by the expansion of road and rail networks to satisfy food and non-food urban needs (Estrada, 2013; Estrada et al., 2017). Although cities concentrate poverty, they also are places of innovation, knowledge, technical expertise, and leadership (van Ginkel, 2008) offering important decision-making tools for primate conservation. For example, green (environmentally friendly) policy initiatives such as recycling, desalination and water treatment, a commitment to re-useable energy, and others can limit a cities ecological footprint (Butler & Laurance, 2008). These policy changes offer the opportunity for these four countries to take advantage of the movement of people from rural to urban areas to reinvest in forest recovery and habitat restoration in these newly vacant spaces (Ashraf, Pandey & de Jong, 2016), and thereby promote conservation policies favouring primate population recovery and expansion.

Table 4 Land area, 2016 human population size, population density, and population growth rates in Brazil, Madagascar, Indonesia, and DRC.

	Brazil	Madagascar	Indonesia	DRC	
Land area km2	8,515,767	587,041	1,904,569	2,344,858	
2016 Population	207,852,865	25,566,097	263,354,770	80,071,935	
2016 Population in urban areas	82%	34%	52%	39%	
2016 Density (persons/km2)	25	44	145	36	
2016 Population growth rate (%) FAO	0.77	2.75	1.07	3.09	
2016 Population growth rate (%) World Bank	0.82	2.69	1.14	3.28	
Note:

Source: FAOStats, http://www.fao.org/faostat/en; the World Bank, http://data.worldbank.org/data-catalog/world-development-indicators (accessed 5 February 2018).

Figure 5 Total urban and rural population growth and projections for (A) Brazil, (B) DRC, (C) Madagascar, and (D) Indonesia.

Steep growth is forecasted for the next few decades with urban populations significantly increasing, while rural populations are expected to decline. Source: http://www.fao.org/faostat/en/#data (accessed 15 August 2017).

Socioeconomic Indicators and Human Development

Gross domestic product per capita

Effective and long-term primate conservation requires economic resources, adequate conservation policies, effective law enforcement, conservation-oriented research, and public interest. If high levels of poverty are predominant, country-wide primate conservation will be a low national priority. The 2015 Gross Domestic Product Per Capita (GDPPC) of Brazil, Madagascar, Indonesia and DRC, was, on average, lower than the world’s average ($10,130) and significantly lower than the average GDPPC for the top 25 most developed nations ($57,509). Among the four countries, DRC and Madagascar have the lowest 2015 GDPPC values ($452 and $402, respectively; Indonesia $3,346; Brazil $8,678) (Table S5). Changes in the GDPPC from 1990 to 2015 for these four primate-richest countries indicate major gains for Indonesia and Brazil whereas the GDPPC has remained very low in DRC and Madagascar (Fig. 6). This is consistent with levels of child malnutrition. The percent of children who are underweight in Brazil is 3.7% (2002), in Indonesia 19.9% (2013), in DRC 23.4% (2013) and in Madagascar 36.8% (2004). In contrast the values for high income countries is 0.9% (2016) (World Bank, 2017).

Figure 6 (A) Gross Domestic Product per capita (GDPPC International USD) in the four countries for the period 1990 to 2015. Included for comparison are the world’s average and the average for the top 25 most developed nations. (B) Trends for DRC and Madagascar. (C) Percent gain for each country for 1990–2006. Available at http://data.worldbank.org/indicator/NY.GDP.PCAP.CD?contextual=max&locations=BR&year_high_desc=false; http://data.worldbank.org/indicator/NY.GDP.PCAP.CD (accessed November 2017).

Human development

The 2015 UN Human Development Index (HDI; a combination of life expectancy, school enrollment, literacy, and income, with the Lowest human development = 0; Highest = 1.0; United Nations Development Programme (UNDP); http://www.undp.org/content/undp/en/home.html) indicates that DRC and Madagascar have the lowest values among the four countries, while the HDI values for Brazil and Indonesia approach the world’s average (Fig. 7). In general, the HDI increased in all four countries from 1990 to 2015, but while the HDI increase in Brazil and Indonesia paralleled increases in the world’s average, human development remained relatively stagnant for Madagascar and DRC (Fig. 7). Values of the HDI for these four countries are, nonetheless, quite low compared to those highest ranking 25 countries worldwide (Fig. 7; Table S5). Low levels of HDI are commonly associated with political instability, extreme income inequality, and limited environmental protection (Alsamawi et al., 2014; Neumayer, 2017). While these four primate-rich countries have much to achieve in human development compared to the top 25 developed nations, it also is clear that the economic standing and human development of Brazil and Indonesia are following a trajectory that is different from that of DRC and Madagascar (Fig. 7). These latter two countries face more serious challenges in securing resources for their human population and for primate conservation.

Figure 7 The 1990–2015 Human Development Index (HDI) in Brazil, Indonesia, Madagascar, and DRC (Lowest human development = 0; highest = 1.0). Also shown is the average HDI for the world and for the top 25 most developed nations.

The number in parentheses after each country indicates their HDI world rank. The number in parenthesis after the name of each country indicates its HDI ranking compared to 188 countries. No data are available for Madagascar for 1990. Source: United Nations Development Program (http://hdr.undp.org/en/composite/trend (accessed 11 January 2018).

Civil conflict

Civil unrest and conflict also affect primate survivorship due to indiscriminate bombing, the spread of toxic chemicals (Douglas & Alie, 2014; Loucks et al., 2009), increases in the availability of firearms, and the increase in bushmeat hunting by soldiers and displaced civilians. Poaching of many primates including gray-cheeked mangabeys (Lophocebus albigena), bonobos and Grauer’s gorillas, for example, has increased markedly in DRC because of ongoing civil wars (Douglas & Alie, 2014; IUCN & ICCN, 2012; McNeely, 2003; Plumptre et al., 2016a). Landmines, the legacy of wars in the 1960’s, 1970’s, and 1990’s, and numerous militia groups continue to jeopardize monkeys and apes in DRC, where civil conflict has interrupted wildlife protection by guards in national parks (e.g., Virunga; Kalpers et al., 2003; McNeely, 2003). Currently, heavily armed militias in the Kasai District, North Kivu and South Kivu in DRC fight for ethnic and political control and, together with illegal miners, prospect for “conflict minerals” (e.g., coltan, tin, tantalum, tungsten, and gold) and diamonds, and hunt primates as bushmeat (Gavin, 2017; Nellemann, Redmond & Refisch, 2010). Similarly, border conflicts between Indonesia and Malaysia on the island of Borneo have caused damage to the forest and wildlife. In the 1990s, however, business and military leaders colluded to suspend conflict in order to cut down and burn millions of hectares of forest to plant cash crops (McNeely, 2003), impacting the survival of entire primate communities. Civil conflict also alters land use patterns and can lead to increased unregulated forest conversion. In the north Sumatran region of Aceh, for example, human conflicts combined with forest fires and legal and illegal logging led to a reduction in forest cover of greater than 30% from 1990 to 2010 (Margono et al., 2012). Disputes over land rights, private corporate actions, and governmental regulations also have led to forest burning and land-clearing across the island of Sumatra, directly threatening the Sumatran langur, banded langur (P. femoralis), and Thomas’s langur (P. thomasi), as well as Bornean orangutans and Müller’s gibbons in Indonesian Borneo (Lanjouw, 2014; Meijaard & Nijman, 2000; Supriatna et al., 2017).

Civil unrest, inter-country wars and continued militarization contribute to the displacement of the local human population, increasing poverty, social insecurity, and environmental damage. The 2017 GPI (http://economicsandpeace.org/), which measures ongoing domestic and international conflict (ODIC), societal safety and security and militarization (IEP, 2017) rank DRC as having the highest values among the four countries (Table 5). Madagascar and Indonesia have lower GPI values for all three insecurity measures, and Brazil has a low value for just one measure, ODIC. When a country’s economic, political and human resources are drained to deal with ongoing civil and ethnic conflicts and societal safety, primate conservation is not a priority. Insecurity and lack of personal safety in these countries are enhanced by prevailing corruption and low-quality governance (see below).

Table 5 The Global Peace Index ranking.

Country	ODIC rank	SSS	MILIT	
Brazil (8th economy)	17	116	109	
DRC (90th economy)	153	127	107	
Madagascar (134th economy)	68	42	23	
Indonesia (15th economy)	92	44	14	
Notes:

Ranking based on the values of the GPI of 163 countries. High values = A higher ranking represents a more unfavorable condition for the three dimensions of the GPI. Sources: Global Peace Index http://economicsandpeace.org (accessed 10 October 2017); economic ranking: World Economic Outlook Database (https://www.imf.org/external/pubs/ft/weo/2017/01/weodata/index.aspx) (accessed 11 October 2017).

ODIC, Ongoing Domestic and International Conflict; SSS, Societal Safety and Security; MILIT Militarization.

Corruption, governance quality and primate conservation

Corruption is a major threat to primates because it distorts environmental laws, giving way to deforestation and land speculation and promoting poverty and illegal activities, including mining, poaching, logging, and the primate trade. Corruption and inequality interact by generating a vicious circle of greed, the unequal distribution of power in society, and the unequal distribution of wealth. The 2016 Transparency International CPI (0: highly corrupt to 100: very clean) profiling 176 countries (Transparency International, 2016) places Brazil with a score of 40 (rank 79), Indonesia a score 37 (rank 90), Madagascar a score 26 (rank 145), and DRC with a score of 21 (rank 156), consistent with the high levels of corruption present in all four countries, but especially in Madagascar, DRC (Transparency International, 2016), and most recently in Brazil. Corruption hampers efforts directed at wildlife conservation and weakens protected area capacity to prevent drivers of primate habitat loss and local species extirpation (see Text S1 for the case of Brazil). In the four countries, laws are often skirted or ignored through bribery and extortion. For example, trading orangutans in Indonesia is a crime but 440 confiscations in the last 25 years have led to only seven convictions and sentencing was lenient (Nijman, 2017). DRC has a patronage system in which the profits of “unofficial economic activities” or “predation” flow upwards to the top of the chain of command hampering the way forward with environmental issues (Baaz & Olsson, 2011: see Text 1). In Madagascar, illegal exploitation and export of rosewood in protected areas, with associated negative effects on wildlife populations, has been facilitated by political instability and corruption (Gore, Ratsimbazafy & Lute, 2013; Randriamalala & Liu, 2010; Schwitzer et al., 2014). Complicity between businesses and politicians had led to the theft of billions of dollars in revenue from national economies, benefitting the very few at the expense of the many and preventing sustainable development (Baaz & Olsson, 2011; Transparency International, 2016). Profiling four key World Bank indicators of governance quality in 2016 indicates that these primate-richest countries all rank significantly lower than the average values for 35 high-income countries (Fig. 8). Overall, weak governance appears to be characteristic of these four countries, with DRC (coded in the World Bank database as Congo Dem. Rep.) and Madagascar ranking lowest (see Freudenberger, 2010). Given high levels of corruption and prevalent low human development, country-wide conservation of primate habitats and populations in these four countries remains a complex challenge. Moreover, measurements of the effectiveness of governance require a thorough causal analysis (with counterfactuals) to determine the degree to which the current status of individual primate species is best attributed to good policies that are poorly implemented, the continuation of ineffective policies, or the result of strong and effectively managed policies (see Baylis et al., 2016).

Figure 8 The graph, produced using the World Bank database, shows the percentile rank of four key World Bank governance indicators for Brazil, DRC, Madagascar, and Indonesia. Percentile rank: the percentage of countries that rate below the selected country.

Higher values indicate better governance ratings. Shown for comparison is the percentile rank for high-income OECD countries (n = 35; Organization for Economic Co-operation and Development). Percentile ranks have been adjusted to account for changes over time in the set of countries covered by the governance indicators. The statistically likely range of the governance indicator is shown as a thin black line. For instance, a bar of length 75% with the thin black lines extending from 60% to 85% has the following interpretation: an estimated 75% of the countries rate worse and an estimated 25% of the countries rate better than the country of choice. Source: http://info.worldbank.org/governance/wgi/index.aspx#reports (accessed 17 November 2017).

Landscape Approaches to Primate Conservation

Protected areas

Protected areas represent an effective conservation tool in which local, state, and national governments can act to protect ecosystems and provide resources to conserve animal populations, provided that these areas also contribute to alleviate rural poverty (Adams & Hutton, 2007). An Africa-wide assessment of which factors were most effective in maintaining great ape populations concluded, after examining 120 areas, that effective law enforcement was the most important (Tranquilli et al., 2012) followed by long-term conservation NGO involvement. Similarly, a recent rangewide assessment of the two great ape taxa in Western Equatorial Africa shows that the presence of wildlife guards was one of the most effective predictors of great ape density (Strindberg et al., 2018), and that intact forest and low human pressure metrics were also key—both of which are generally characteristic of the protected areas and selectively-logged Forest Stewardship Council (FSC)-certified concessions of Central Africa. Globally, protected area networks are located in ecological zones that have low value and low demand for land conversion, are inexpensive to protect, and, some, but by no means all, are, located far from areas of high biodiversity (Joppa & Pfaff, 2009). As a result, this discrepancy or this mode of selection has placed primate-rich lowland forests at risk because lowland forests offer profitable opportunities to obtain land well-suited to industrial agriculture (Venter et al., 2014) or clear-cutting for timber. In this regard, governments need to partner with the scientific community and the expertise of local, regional, national, and international NGOs to design extensive networks of protected areas and private reserves that have as their goal the creation of ecological zones and land use policies that collectively sustain both biodiversity and human communities (Hill et al., 2015). There is evidence that protected areas provide sustainable core habitat for primates. They represent a keystone tool for the conservation of threatened primates in Brazil’s Atlantic forest. For example, almost 80% of the total localities of Atlantic Forest where muriquis (Brachyteles spp.) presently inhabit are protected areas (private or governmental—state and federal units, Strier et al., 2017). In Central Africa, a long-term study (2007–2014) in which camera traps were used to census terrestrial mammals found strong evidence of stability in several threatened African primates such as the l’Hoest’s monkey, mandrills (Mandrillus sphinx) and chimpanzees (Beaudrot et al., 2016).

Conservation efforts targeted to deliberately increase positive human influences, including veterinary care and close monitoring of individual animals succeeded in doubling the Virunga mountain gorilla population over 40 years (Robbins et al., 2011). These gorillas occur in protected areas, including in DRC. Protected areas are effective in minimizing population decline as has been reported for the pale-thighed langur (Presbytis siamensis) in Sumatra and the red-fronted brown lemur (Eulemur rufifrons) in Madagascar (Beaudrot et al., 2016). From 1990 to 2000, protected areas in Sumatra experienced lower deforestation rates than nearby unprotected areas (Gaveau et al., 2009; Gaveau, Wich & Marshall, 2016). In Zanzibar, Tanzania, mean group sizes of the Zanzibar red colobus Piliocolobus kirkii were significantly higher in protected areas (21 individuals) than outside protected areas (13 individuals). Clearly, individuals outside of protected areas are at greatest risk (Davenport et al., 2017). In this regard, Brazil has 29% of its land under protection, DRC 13%, Madagascar 12%, and Indonesia 12% (Table 6; see Text S1 for additional information).

Table 6 The number and accumulated extent of protected areas in Brazil, Madagascar, Indonesia, and DRC.

	Brazil	Madagascar	Indonesia	DRC	
Protected areas	2,190	221	646	90	
km2 protected	2,468,479	71,000	226,249	260,000	
Land area km2	8,515,767	587,041	1,904,569	2,344,858	
% of land area protected	29	12	12	13	

Assessing the overlap between protected areas and primate distributions

Modeling the distribution of protected areas and primate distributions in the four countries showed that, on average, primates in Brazil have 38% of their range included within protected areas; 38% in Madagascar, 17% in Indonesia, and 14% in DRC, suggesting that the great majority of primate populations exist outside of protected areas (Fig. 9; Fig. S7; see Strindberg et al., 2018 for the case of central African chimpanzees and western lowland gorillas in Western Equatorial Africa where 80% of both primates occur outside of protected areas). Regrettably, the distribution of protected areas in each of the four countries is extremely patchy, and in many cases subpopulations of the same species are isolated from each other and inhabit areas that are experiencing considerable deforestation and fragmentation as they are increasingly impacted by agricultural expansion, logging, and illegal hunting as well as an ever-growing urban footprint (Figs. 3 and 9) (Gouveia et al., 2017; Mascia et al., 2014; Rovero et al., 2015; Spracklen et al., 2015; Waeber et al., 2016). Due to illegal activity in the Brazilian Amazon, natural resource reduction is pervasive. Most transgressions were related to habitat degradation (37%), illegal fishing (27%), and game hunting (18%) (Kauano, Silva & Michalski, 2017). Increasing human population density within 50 km of a protected area is a crucial factor that promotes illegal activities. Meeting global goals for protected-area coverage will be insufficient to protect biodiversity unless these areas are well managed and properly located (Butchart et al., 2015). Analysis of the distribution of protected areas and primate distributions is critical for diagnosing areas in need of protection. For example, whereas 22% of the distribution of the Bornean orangutan is in protected areas and 29% occurs in forest concessions, the remaining 49% is in unprotected and commercially developed forests (Wich et al., 2012). A similar pattern emerged in an earlier analysis of all primate species in Indonesian Borneo (Meijaard & Nijman, 2003).

Figure 9 Distribution of protected areas and primate distributions in (A) Brazil, (B) DRC, (C) Madagascar, and (D) Indonesia.

In this model, primate species distributions are based on data from the IUCN Red List (accessed May 2017), protected areas distributions from UNEP-WCMC (2017) and forest cover from Hansen et al. (2013). Images are scaled to ca. 300 m of spatial resolution. We included 2,190 protected areas in the Brazil dataset, 49 in DRC, 147 in Madagascar and 646 in Indonesia (Text S1).

Community forest management, habitat restoration and landscape connectivity

Community forest management (CFM) aims to reduce deforestation and maintain biodiversity while also improving local human welfare (alleviate poverty). In general, there is evidence of CFM being associated with greater tree density and basal area (Bowler et al., 2012). A review of 33 community forests (all but one in Latin America, the other in India) showed that a commitment to land-sharing (combining forms of agroforestry along with forest managed by local communities in which resources are extracted sustainably) can lead to reduced rates of deforestation compared to protected forests (Porter-Bolland et al., 2012). In another study of CFM certification of timber, based on 318 comparisons from 50 studies distributed across Africa, Asia, and South and Central America, CFM performed better than open access areas in 56% of 185 comparisons, equally in 25% and worse in 19% (comparisons focused on economic, social and environmental variables) (Burivalova et al., 2017; Dasgupta & Burivalova, 2017; Ndoye & Chupezi Tieguhong, 2004). Similarly, a nation-wide survey in Madagascar of CFM impacts on household living standards (as measured by per capita consumption expenditures) showed that well-being was stronger for households closer to forests and households with more years of education (Rasolofoson et al., 2017). In another study in Madagascar, CMF was shown to reduce deforestation in CFM localities that do not permit commercial uses of wood compared to areas that lack CFM or in CFM areas that allow commercial uses (Rasolofoson et al., 2015). In Indonesia, the total area of CFM forests (Hutan Desa, or village forest, is an approach that stresses local village governance and autonomy in forest protection and in controlling resource extraction by outside groups) increased from 750 km2 in 2012 to 2500 km2 in 2016. A spatial matching approach showed that under a Hutan Desa management scheme, deforestation was avoided compared to the expected likelihood of deforestation in the absence of Hutan Desa management (Santika et al., 2017b).

Forests are one of the few resources accessible to local communities in primate range countries, and participating in their ownership, stewardship, and restoration can provide food, economic opportunity, and income to poor people (Bakwaye Flavien et al., 2016; Porter-Bolland et al., 2012). Reforestation is an important conservation tool to help both rural communities and to mitigate species extinction due to habitat loss, fragmentation, and isolation, especially if it involves protecting large forest areas (Taubert et al., 2018). An expansion in available habitat via restoration can facilitate an increase in species’ population size and connect fragments and protected areas, if strategically located restored forest can promote immigration and gene flow from previously isolated but now source populations (Hylander & Ehrlén, 2013). Targeting habitat restoration to areas of once contiguous forest using corridors 1-km wide between the most extensive, intact, and closest forest fragments can have a positive effect on wildlife population expansion (Newmark et al., 2017). A study in the Atlantic Forest of Brazil that modeled the use of forest corridors as a conservation tool found that regenerating corridors totaling 6.4 K ha would result in a continuous forested area measuring 251.9 K ha. Although full regeneration of these corridors is likely to take 10–40 years (Newmark et al., 2017), extinction-prone primate species such as golden lion tamarins (Leontopithecus rosalia) and golden-headed lion tamarins (Leontopithecus chrysomelas) can disperse through linked forests that are <10 years old (Dosen, Raboy & Fortib, 2017; Newmark et al., 2017). Landscape connectivity also can include community managed forests in which agroecosystems such as shade-grown coffee (Coffea spp.), cacao (Theobroma cacao), and cardamom (Elettaria cardamomum), as well as small shaded mixed plantations of natural rubber (Hevea brasiliensis) and oil palm, among other arboreal crops, provide income for farmers and temporary habitat, food resources, and dispersal routes for isolated segments of primate subpopulations (Estrada, Raboy & Oliveira, 2012; McLennan, Spagnoletti & Hockings, 2017). Still, the persistence of primates in agroecosystems in Brazil, Madagascar, Indonesia and DRC may not be a long-term sustainable conservation solution (Text S1).

Primate rewilding

Where primate species are locally extirpated, reintroductions may be a feasible conservation strategy if there is long-term protection of forests and monitoring of population changes (Kierulff et al., 2012; Beck et al., 2007; Wilson et al., 2014). In general, guidelines for most species, including great apes, underline the importance of ensuring that the threat that caused the animals to become locally extinct (such as poaching) has ceased before attempting reintroduction (Beck et al., 2007; IUCN/SSC, 2013). Reintroduction and translocation programs also serve to intensify public interest on conservation issues, especially when combined with social media (Kierulff et al., 2012). Reintroduced primates include orangutans and slow lorises in Indonesia (Banes, Galdikas & Vigilant, 2016; Moore, Wihermanto & Nekaris, 2014; Wilson et al., 2014), and golden lion tamarins, pygmy marmosets (Cebuella pygmaea) and northern muriquis in Brazil (Car, Queirogas & Pedersoli, 2015; Kierulff et al., 2012; Melo, 2016; Ruiz-Miranda et al., 2006). Some of these releases, e.g., golden lion tamarins, led to the establishment of self-sustaining populations, whereas in others, for example, Javan slow lorises (Nycticebus javanicus), high mortality in the first few months, questions the viability of these programs (Moore, Wihermanto & Nekaris, 2014). In Madagascar, there have been reintroductions and translocations of captive-born and wild-born lemurs (Schwitzer et al., 2013). This has resulted in successful population establishment in the cases of released aye-ayes (Daubentonia madagascariensis), captive-bred black-and-white ruffed and collared-brown lemurs (Eulemur collaris), but in several instances, there was high mortality due to natural predation (Britt, Welch & Katz, 2004; Donati et al., 2007; Mittermeier et al., 2010). In contrast, the translocation of black-and-white ruffed lemurs and diademed sifakas from a forest selected for clearing by a mining company to the nearby Analamazaotra Special Reserve (ASR), was successful (Day et al., 2009). After several years of rehabilitation, bonobos rescued from the illegal trade also have been successfully reintroduced in the “Ekolo Ya Bonobo” release site in DRC (http://www.lolayabonobo.org/ekolo-ya-bonobo; accessed 30th November 2017). Nevertheless, extreme caution must be taken in evaluating areas for reintroduction, as in many cases humans and primates favor the same ecological zones, and primates may face severe competition from human, as is the case for the Bornean orangutan (Santika et al., 2017a) and the population of black-and-white ruffed lemurs currently living in the ASR (Rasoamanarivo et al., 2015). These two studies may guide future attempts (Text S1). Finally, reintroductions are costly relative to other options (e.g., long-term protection of forested land) and therefore are often best used as a last resort (Wilson et al., 2014).

Socially–oriented conservation actions for averting local extinction threats to primates

The development of community-based local education programs, action groups, and NGOs/Associations that focus on primate conservation initiatives are key instruments that can successfully result in local and long-term conservation of primate species. The involvement of NGOs (International Committees for Conservation and Management—ICCM and the Pro-Muriqui Institute) has been crucial in Brazil for the conservation of threatened species such as the golden lion tamarin and the muriqui (Jerusalinsky, Talebi & Melo, 2011; Rylands et al., 1998; Text S1). The Critically Endangered Javan slow loris is one of the focal species of the Little Fireface Project (http://www.nocturama.org). Implemented in 2011, this project has involved a wide range of audiences and stakeholders, providing annual training sessions for law enforcement officers and coordinated biannual events in villages close to where wild slow lorises occur, to increase protection and pride in this endemic species. A population of these slow lorises has been monitored on Mt Papandayan, West Java, for seven years, revealing vital information on their biology and conservation (Nekaris, 2016; Nekaris et al., 2018; Text S1). In the DRC, international conservation NGOs are working in and around most conservation landscapes, with coordination offices in the capital; some of these NGOs have been working in DRC for over 30 years. NGOs support the wildlife authority of the ICCN (Congolese Institute for Nature Conservation), and provide technical assistance (training, equipment) to government antipoaching patrols, that play a critical conservation role inside national parks, including the development of a system for rapid collection of both patrol and survey data in the field (SMART, http://smartconservationtools.org/) (Text S1).

Literacy is another critical factor in developing effective conservation education programs (Clayton & Myers, 2009; Oboh & Tsue, 2010). Youth (15–24 years) literacy rates are 76.8% (2012 data) in Madagascar, 85.0% in the DRC (2016), 98.8% in Brazil (2014), and 99.7% in Indonesia (2016) (http://data.uis.unesco.org/#). In addition, in rural parts of DRC and Madagascar, adult literacy is some 25% lower than in urban populations (UNESCO, 2006). Educational programs targeted at less literate populations are more effective when environmental messages are presented using simply written phrases, radio and television programs, music, images, live performances, and other non-written forms of communication.

In Madagascar, the NGO Reniala acts to protect forests, rehabilitate lemurs from the pet trade, provide incentives to discourage hunting, and has developed alternative livelihood projects for local residents, such as beekeeping (https://association-reniala.jimdo.com/). Centre Valbio is a research center with an integrated conservation program, that works directly with the Malagasy government in the Ranomafana National Park—41,500 hectares of rainforest that includes the golden bamboo lemur (Hapalemur aureus), discovered at this site in 1986 (http://www.stonybrook.edu/commcms/centre-valbio/conservation.html). The Maromizaha forest Conservation and Community Project in Madagascar protects a large forest tract with 13 species of lemurs, using forest-friendly alternative agricultural practices and promoting the development of ecotourism (Gamba et al., 2013; Neudert, Ganzhorn & Wätzold, 2016). Because of the recent growth in trained primatologists in the four countries, their conservation concerns have led to the creation of professional societies that can more effectively articulate conservation concerns with local governments, NGOs, rural communities and international societies (Text S1). In Madagascar, conservation education, especially of young children, also has made important strides in protecting primates (Dolins et al., 2010).

Conclusions and Key Challenges Ahead

Primate conservation is a global, multilayered, biological, ecological, and social issue. There are over 500 primate species in the wild and these taxa differ in ecological requirements, behavioral flexibility, reproductive capacity, social systems, and are long-lived (Fig. 10). As a result, their responses to conservation initiatives are often difficult to assess in both the short and long term. There is no single blueprint or best course of action for advancing primate conservation in Brazil, Madagascar, Indonesia, and DRC. Each country differs in its history, societal and economic needs, and current environmental and governmental policies that are driving primate habitat loss and population decline. These four countries face unprecedented environmental and social challenges in implementing effective primate conservation (Fig. 11). They have rapidly growing human populations and low human development indices compared with more developed nations. Each has also experienced large-scale losses of native vegetation and other natural resources plus high levels of corruption and weak governance. Each country’s desire to move its economy forward to meet the needs of its population remains a priority but this seems difficult to achieve in a global system in which international trade led by the demands for food and non-food products by a small set of developed and consumer nations distract attention from the needs of their local populations. Despite significant increases in revenues derived from agricultural exports in these four countries, millions of their citizens remain undernourished, undereducated, and poor (World Bank, 2017). While Brazil has made important strides in reducing deforestation, decreasing poverty, and fostering science and education with direct positive impacts on primate conservation, a change in government policies in 2012 reduced the protection of natural vegetation on private lands (Brancalion et al., 2016) and laws governing protected areas were reduced and weakened (Bernard, Penna & Araújo, 2014). Funding for science also was reduced (Overbeck et al., 2018). This has resulted in a sharp increase in deforestation rates (Fig. 2), with expected negative effects on biodiversity, primates, people’s livelihood’s, and conservation.

Figure 10 Photos of selected primates from each country.

Conservation status and photo credits include the following: (A) DRC, Grauer’s gorilla (Gorilla beringei graueri), Critically Endangered, (Photo credit: J. Martin), (B) Madagascar, Sahafary sportive lemur (Lepilemur septentrionalis) Critically Endangered (Photo credit: R. A. Mittermeier), (C) Indonesia, Javan slow loris (Nycticebus javanicus), Critically Endangered (Photo Credit: Andrew Walmsley/Little Fireface Project), (D) Brazil, northern muriqui (Brachyteles hypoxanthus), Critically Endangered (Photo credit: Raphaella Coutinho), (E) Brazil, pygmy marmoset (Cebuella pygmaea), Vulnerable, (Photo credit: Pablo Yépez), (F) Sumatran orangutan (Pongo abelii), Critically Endangered (Photo Credit: Perry van Duijnhoven).

Figure 11 Diagram summarizing key environmental challenges common to Brazil, DRC, Madagascar, and Indonesia that affect conservation of their primate fauna.

The relative importance of some pressures and population aspects vary from country to country. For example, hunting in DRC is a large-scale pressure because the local human population has little or no access to domestic meat. Because of their large size and low population density relative to the size of the country, Brazil and DRC are in a better position to anticipate the direction of these pressures and prevent primate declines and extirpation. However, in contrast to Brazil, DRC is particularly poor, its human population is rapidly growing, and human development is very low, whereas civil unrest is predominant and corruption and weak governance are an ever-present condition. Madagascar differs from these two countries, and from Indonesia in having a very small percentage of its original forest left. A rapidly expanding human population and high levels of poverty and weak governance are predominant. Indonesia is a developing country with a large human population that has embarked on a policy of rapidly replacing its forests with commercial plantations and expanding industrial logging at the expense of biodiversity.

Given the rapid pace and large scale at which native forests have been cleared in Latin America and Indonesia to expand agriculture to satisfy global and local demands (Tilman et al., 2017), critically evaluated approaches are needed to ensure primate survival. For example, it is argued that promoting “sustainable intensification” of agriculture on already cleared lands could readily supply production that might otherwise be expected to come at the cost of future land conversion (Carlson et al., 2018). This requires linking smallholders (farmers) with commercial international agriculture (Goldsmith & Cohn, 2017). This, however, does not mitigate the already high environmental costs of cleared land. Moreover, global dietary changes, promoted in large part by multinational businesses and designed to expand corporate profit margins, will require these primate-rich countries to convert additional forested land into monocultures (Kastner et al., 2012; Tilman & Clark, 2014). This is likely to happen more slowly in DRC, as civil war, political instability, governance issues, and continued poverty (according to the internationally recognized metrics used by the World Bank/UNDP) limit international investment (Kastner et al., 2012). Based on a range of global factors, agribusiness corporations may switch to different crops and land-use patterns to maximize profit or may increase or decrease investments in other countries leading to increased environmental damage and poverty (Carrasco et al., 2014; Lim et al., 2016; Villoria et al., 2013; Weng et al., 2013). Intensification of agriculture to increase yields does not necessarily contribute to global hunger reduction, as an unequal amount of food and nonfood products are used by already well-fed people in a small number of consumer nations. Rather, food security needs to increase in areas of the world where the hungry live using eco-efficient approaches that encourage sustainable productivity and incorporates natural biodiversity, and clean and reusable forms of energy, while sustaining multiple ecosystem services (Keating et al., 2010). Using an eco-friendly approach, agriculture practiced by small landholders and sensitive to local markets and conditions rather than large-scale industrial farming, is the key to food security in the developing world (Runting et al., 2015; Tscharntke et al., 2012).

Clearly, additional research is needed to examine the role of local and global market demands on primate conservation (Larrosa, Carrasco & Milner-Gulland, 2016), including studies to evaluate the extent to which the reduction of land for purposes of agricultural conversion benefits the local human and nonhuman primate communities. Within this framework, economic instruments targeted to consumer nations such as taxes on agricultural inputs and taxes on consumption as well as investment in sustainable agri-environmental production that guarantees the persistence of multiple ecosystem services may be, in some countries, viable alternatives to mitigate the negative impacts of agricultural expansion (Hanspach et al., 2017; Nepstad et al., 2014; Tanentzap et al., 2015).

Worldwide, policies targeting consumer nations that reduce their ecological footprint in primate range regions are needed. Green tagging and certification, greater controls on fair trade, corporate responsibility in recycling, and pollution and carbon emmisions control, the elimination of excessive packaging, and the sustainable purchasing of goods and services are critical tools for lowering worldwide demand for processed materials (Moran, Petersone & Verones, 2016) and would help alleviate pressures on primate habitats (Dalerum, 2014). As part of the “greening” of trade, international corporations should add “environmental” costs to products so that there is a continuous regeneration of funds to sustainably promote conservation (Butler & Laurance, 2008). Alternatively, the World Bank or UN could require that corporations and consumer nations pay into a sustainability/conservation fund based on their levels of consumption and environmental damage (e.g., like a carbon tax; Carbon Tax Center https://www.carbontax.org; consulted August 2017). In countries in which the rural poor depend on forest products, community forest management could bridge or integrate the needs of conservation and commodity production, sustainably safeguarding the continued integrity of complex ecological systems (Sharif & Saha, 2017). The recent environmentally-oriented, demand-side policies regarding illegal timber imports by the EU (EU, 2010), the EU resolution on oil palm production and deforestation (EP, 2017), and the Amsterdam Declaration to eliminate deforestation from agricultural commodity chains (Amsterdam Declaration, 2015) represent important and positive “green” changes that need to be adopted by the U.S., China, and other consume nations. However, the continued growth of the global demand for forest-risk agricultural and nonfood commodities requires additional legislation and a stronger global effort at regulating the negative impact of unsustainable commodity trade (Henders et al., 2018).

In the context of large-scale deforestation, Brazil, Madagascar, Indonesia, and DRC face additional challenges that require cost-effective policies designed to maintain intact areas of forest and biodiversity (Busch & Engelmann, 2018) (Fig. 11). One approach to achieve this goal is the REDD+ program where funds are provided by consumer nations to forest-rich countries to offset emissions from deforestation, forest fragmentation, and other forms of environmental degradation (García-Ulloa & Koh, 2016; Venter & Koh, 2012). These funds could be targeted to expand forested habitats and connect forest fragments, as well as provide security for local populations by increasing the economic and ecological value of maintaining forested land. These programs are just beginning but are showing some promising results in DRC (Fobissie, 2015), in Makira, Madagascar (see https://madagascar.wcs.org/Makira-Carbon.aspx) and in the Atlantic Forest of Brazil (Brancalion et al., 2013).

The forecasted future human population and economic growth of Brazil, Indonesia, DRC, and Madagascar along with increased global and local demands for food and non-food products will heighten pressures on primate populations in these four countries. The Brazilian ability to combat deforestation by 80% between 2005 and 2013 is an example that could be followed by the other three countries (Dobrovolski & Rattis, 2014; Nepstad et al., 2014). Countries that rely on agricultural and natural resource exports from these four primate-rich countries must become major contributors to conservation efforts that take place beyond their borders. The safeguarding of the primate fauna in each country will require an increase in suitable land devoted to protected areas and improved conservation management, as many species lack adequate protection (Joppa & Pfaff, 2009). In addition, given that the unprecedented globalized demand for illegal wildlife, the bushmeat trade, and the use of body parts in traditional medicine and as trophies, is rapidly depleting natural primate populations (Bennett, 2002; UNODC, 2016), an international agency, such as Interpol, that has the capacity to conduct and coordinate counter intelligence investigations worldwide is critically needed. These international investigations can identify criminal organizations involved in the illegal trade, which should be considered a form of bioterrorism (Figs. 4 and 11). Given the severity of this problem, stopping the supply chain of illegal primate hunting and primate trade in the four countries needs to be included in integrated conservation models (Brashares et al., 2014) that also addresses government corruption (Estrada et al., 2017). This also requires a focused effort to promote a positive attitude, both in primate range countries and in consumer nations towards environmental protection and conservation education, and interventions need to provide resources and access to information to encourage members of local communities to protect wildlife (Challender & MacMillan, 2014).

Our review has shown that local and global social, economic and political factors imperil the persistence of primate populations in Brazil, Madagascar, Indonesia, and DRC, and that more needs to be done by local governments and international bodies to ensure that primates, a critical component of each nation’s natural heritage and biodiversity, do not become rare, locally extirpated, or in the case of endemic species, extinct. If this is allowed, these countries risk losing complex ecosystem services and social, historical, and cultural relationships that have persisted between human primates and nonhuman primates over many thousands of years (Chandra, 2017; Fuentes, 2012; Voigt et al., 2018). These countries also risk the destabilizing consequences of habitat degradation, pollution, climate change, and food insecurity for their human populations Because Brazil, Madagascar, Indonesia, and the DRC harbor 65% of the world’s primate species, these countries are of critical significance for global primate conservation. Consequently, urgent local and global action must be taken to reverse the current situation of impending primate extinctions.

Supplemental Information

Supplemental Information 1 Fig. S1. Expansion of agricultural land for the period 2001 to 2015 in Brazil, Indonesia, Madagascar and DRC.

Available at FAOStats http://www.fao.org/faostat/en/#compare (accessed 10 February 2018). See Text S1 for limitations of the FAO data.

Click here for additional data file.

Supplemental Information 2 Fig. S2. Trends in the growth of cattle populations and in the production of some of the most important agricultural commodities in Brazil.

Available at http://www.fao.org/faostat/en/#data (accessed 14 February 2018; for a definition of the category Roundwood nonconiferous see http://www.fao.org/waicent/faostat/forestry/products.htm#S2; http://www.fao.org/faostat/en/#compare (crops processed). The category Roundwood-tropical has no data available in FAO data. See Text S1 for limitations of the FAO data.

Click here for additional data file.

Supplemental Information 3 Fig. S3. Trends in the growth of the cultivated areas devoted to roots and tubers, maize, and rice paddy production and in two important arboreal food crops in DRC.

Also shown is the growth trend in the harvest of hardwoods. Available at http://www.fao.org/faostat/en/#compare (crops processed) (accessed 1 April 2017). Note: starting year may differ from one crop to another due to the lack of data available for those years in the FAO databases. See Text S1 for limitations of the FAO data.

Click here for additional data file.

Supplemental Information 4 Fig. S4. Trends in the growth of cultivated areas including roots and tubers, maize, and rice paddy production in Madagascar. Also shown is the growth in the extraction of hardwoods.

Available at http://www.fao.org/faostat/en/#compare (crops processed) (accessed 1 April 2017). See Text S1 for limitations of the FAO data.

Click here for additional data file.

Supplemental Information 5 Fig. S5. Trends in the area devoted to the cultivation of rice, oil palm, natural rubber and the increase in the volume of roundwood extraction in Indonesia.

Available at http://www.fao.org/faostat/en/#compare (crops processed) (accessed 5 April 2017). See Text S1 for limitations of the FAO data.

Click here for additional data file.

Supplemental Information 6 Fig. S6. Optimistic, business as usual, and pessimistic scenarios of expected spatial conflict between agricultural expansion and primate distributions in the 21st century in Brazil, DRC, Madagascar and Indonesia.

The table at the bottom shows the predicted agricultural expansion values (%) to take place by 2050 and 2100 under each of the three scenarios. Notice the spatial shift of conservation conflicts in the pessimistic models, with Madagascar and DRC reducing agricultural expansion by 2100. This is based on the expectation of the abandonment of some agricultural areas, by 2100 in DRC and Madagascar. This condition, however, may not imply an immediate benefit for primates and other species, as the areas would have been over-exploited prior to abandonment and unlikely to regenerate back to natural forest.

Click here for additional data file.

Supplemental Information 7 Fig. S7. Frequency distribution of the area of the ranges of primate species (blue) and the area of their ranges that overlap with protected areas.

(A) Brazil, (B) DRC, (C) Madagascar and (D) Indonesia.

Click here for additional data file.

Supplemental Information 8 Table S1. Biological richness of four major vertebrate groups in Brazil, DRC, Madagascar and Indonesia.

Source: IUCN, 2017 http://www.iucnredlist.org–consulted August 2017.

Click here for additional data file.

Supplemental Information 9 Table S2. Number of primate species, genera and families currently present in Brazil, DRC, Madagascar and Indonesia.

Also shown is the number of species threatened and with declining populations. Source of data: IUCN, 2017 http://www.iucnredlist.org (consulted February 13th, 2018). Three families are shared by DRC and Indonesia: Lorisidae, Cercopithecidae and Hominidae. No primate species are shared by these four countries.

Click here for additional data file.

Supplemental Information 10 Table S3. Tree cover loss (>30% canopy cover) for the period 2001 to 2016.

Source: Global Forest Watch (http://www.globalforestwatch.org (accessed 11 January 2018). All areas are in ha.

Click here for additional data file.

Supplemental Information 11 Table S4. Expansion estimates of agricultural land in Brazil, Indonesia, Madagascar and DRC for the period 2001 to 2015.

Also shown is agricultural land as percent of the country’s land area. Source of data: FAOStats http://www.fao.org/faostat/en/#data (accessed 12 February 2018). Increases or decreases from year to year can be calculated by subtracting values between years. See Text S1 for limitations of the FAO data sets.

Click here for additional data file.

Supplemental Information 12 Table S5. Gross Domestic Product Per Capita (GDPPC) and the Human Development Index for the 25 most developed nations in the world and for Brazil, DRC, Madagascar and Indonesia.

Source HDI: http://hdr.undp.org/en/countries/profiles/COD (accessed 5 February 2018) Source GDPPC: http://data.worldbank.org/indicator/NY.GDP.PCAP.CD?contextual=max&locations=BR&year_high_desc=false; http://data.worldbank.org/indicator/NY.GDP.PCAP.CD (accessed 5 February 2018).

Click here for additional data file.

Supplemental Information 13 Supplementary text (Text S1).

Click here for additional data file.

Paul A. Garber is forever grateful to Jennifer A. Garber for inspiring him to redirect his efforts to protecting the world’s threatened primate populations. Alejandro Estrada is thankful to Erika and Alex for always supporting my interests in primate reaearch and conservation. Alejandro Estrada would like to thank P.A. Garber and all our couathors for their collaboration to move this article forward.

Additional Information and Declarations

Competing Interests

Author Contributions

Data Availability

Russell A. Mittermeier and Anthony B. Rylands are employed by Global Wildlife Conservation, Christoph Schwitzer is employed by Bristol Zoological Society, Christian Roos is employed by Deutsches Primatenzentrum, Made Wedana is employed by The Aspinall Foundation and Arif Setiawan is employed by SwaraOwa.

Alejandro Estrada conceived and designed the experiments, performed the experiments, analyzed the data, contributed reagents/materials/analysis tools, prepared figures and/or tables, authored or reviewed drafts of the paper, approved the final draft.

Paul A. Garber conceived and designed the experiments, performed the experiments, analyzed the data, contributed reagents/materials/analysis tools, prepared figures and/or tables, authored or reviewed drafts of the paper, approved the final draft.

Russell A. Mittermeier performed the experiments, analyzed the data, contributed reagents/materials/analysis tools, approved the final draft.

Serge Wich performed the experiments, analyzed the data, contributed reagents/materials/analysis tools, authored or reviewed drafts of the paper, approved the final draft.

Sidney Gouveia performed the experiments, analyzed the data, contributed reagents/materials/analysis tools, prepared figures and/or tables, authored or reviewed drafts of the paper, approved the final draft.

Ricardo Dobrovolski performed the experiments, analyzed the data, contributed reagents/materials/analysis tools, prepared figures and/or tables, authored or reviewed drafts of the paper, approved the final draft.

K.A.I. Nekaris performed the experiments, analyzed the data, contributed reagents/materials/analysis tools, prepared figures and/or tables, authored or reviewed drafts of the paper, approved the final draft.

Vincent Nijman performed the experiments, analyzed the data, contributed reagents/materials/analysis tools, prepared figures and/or tables, authored or reviewed drafts of the paper, approved the final draft.

Anthony B. Rylands conceived and designed the experiments, performed the experiments, analyzed the data, contributed reagents/materials/analysis tools, authored or reviewed drafts of the paper, approved the final draft.

Fiona Maisels performed the experiments, analyzed the data, contributed reagents/materials/analysis tools, prepared figures and/or tables, authored or reviewed drafts of the paper, approved the final draft.

Elizabeth A. Williamson performed the experiments, analyzed the data, contributed reagents/materials/analysis tools, prepared figures and/or tables, authored or reviewed drafts of the paper, approved the final draft.

Julio Bicca-Marques performed the experiments, analyzed the data, contributed reagents/materials/analysis tools, authored or reviewed drafts of the paper, approved the final draft.

Agustin Fuentes performed the experiments, analyzed the data, contributed reagents/materials/analysis tools, approved the final draft.

Leandro Jerusalinsky performed the experiments, analyzed the data, contributed reagents/materials/analysis tools, approved the final draft.

Steig Johnson performed the experiments, analyzed the data, contributed reagents/materials/analysis tools, authored or reviewed drafts of the paper, approved the final draft.

Fabiano Rodrigues de Melo performed the experiments, analyzed the data, contributed reagents/materials/analysis tools, approved the final draft.

Leonardo Oliveira performed the experiments, analyzed the data, contributed reagents/materials/analysis tools, approved the final draft.

Christoph Schwitzer performed the experiments, analyzed the data, contributed reagents/materials/analysis tools, authored or reviewed drafts of the paper, approved the final draft.

Christian Roos performed the experiments, analyzed the data, contributed reagents/materials/analysis tools, authored or reviewed drafts of the paper, approved the final draft.

Susan M. Cheyne performed the experiments, analyzed the data, contributed reagents/materials/analysis tools, approved the final draft.

Maria Cecilia Martins Kierulff performed the experiments, analyzed the data, contributed reagents/materials/analysis tools, authored or reviewed drafts of the paper, approved the final draft.

Brigitte Raharivololona performed the experiments, analyzed the data, contributed reagents/materials/analysis tools, authored or reviewed drafts of the paper, approved the final draft.

Mauricio Talebi performed the experiments, analyzed the data, contributed reagents/materials/analysis tools, approved the final draft.

Jonah Ratsimbazafy performed the experiments, analyzed the data, contributed reagents/materials/analysis tools, approved the final draft.

Jatna Supriatna performed the experiments, analyzed the data, contributed reagents/materials/analysis tools, approved the final draft.

Ramesh Boonratana performed the experiments, analyzed the data, contributed reagents/materials/analysis tools, approved the final draft.

Made Wedana performed the experiments, analyzed the data, contributed reagents/materials/analysis tools, approved the final draft.

Arif Setiawan performed the experiments, analyzed the data, contributed reagents/materials/analysis tools, approved the final draft.

The following information was supplied regarding data availability:

The research in this article did not generate any data or code.

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
