# Peer review of "Primates in peril: the significance of Brazil, Madagascar, Indonesia and the Democratic Republic of the Congo for global primate conservation"

_PeerJ, doi:10.7717/peerj.4869_

## Round 0.1 · original submission · Minor Revisions

The paper is well written and very thorough in covering four key countries that contain most of the world’s remaining primates. While heavier on threats to primates and lighter on solutions, understanding the complex challenges to conservation in each area is of great value going forward.

The only main concern I have is the lack of discussion of data sources. I realize the paper is complex and involves a tremendous amount of work. Using what data are available is a good approach but only in one place is there a caveat placed on the data (that I noted) and that is regarding the self-reporting from countries on FAO. One of the reviewers flags this issue as well. A revision should include a discussion of the data sources and evaluation of the strengths and weaknesses. This critical evaluation will provide those engaged in allied conservation and development work a starting place for understand what data need to be collected yet and how to interpret the findings of the review relative to the data sources. In addition to the critical evaluation, please note the additional points raised by each reviewer.

Overall, the paper is very strong and worthy of publication.

This doesn’t have to be included but a recent paper discusses the relationship between male investment in reproduction and increased rate of species’ extinction—might be of interest in the final sections in looking at species at particular risk: Martins MJF, Puckett TM, Lockwood R, Swaddle JP, Hunt G. High male sexual investment as a driver of extinction in fossil ostracods. Nature. 2018;10.1038/s41586-018-0020-7. doi: 10.1038/s41586-018-0020-7.

Reviewer 1 ·

Basic reporting

This paper represents an excellent review of the current situation that leads to the endangerment of the primates in Brazil, Democratic Republic of Congo (DRC), Madagascar, and Indonesia. It primarily relies on presenting current statistics of the situation, which is extremely valuable. This will be very useful to conservation biologists and individuals seeking to set conservation policy.

I have few comments as there seems little point in addressing or questioning specific statistics; however, I would like to see 2 major issues addressed
First, the authors rarely question the validity or strength of the data that they are relying on to make there arguments, and while I do not see any current alternatives to use these data, I think that pointing out where data is weak would help researchers in the future as they could focus on strengthening these areas. I see two particular areas. The IUCN species categorizations area typically all we have to assess endangerment, but in some places (particularly the DRC) these estimates have to be considered poor at best and improved efforts to survey over multiple time periods are desperately needed. Second, we know frightening little about how tropical forests and primates will respond to climate change and this relies on long-term research which is extremely hard to fund and more and more long-term primate sites are closing.
Second, the authors make many many claims of what needs to be done, but almost totally shy away from stating the likelihood that these challenges will be met. This is particularly the case in the conclusions. If these challenges cannot likely be met (and to me this is likely often to be the case), the authors should make the appropriate speculation that they will not be met – the authors are in a good situation to make these statements, or at least better than most readers. Without doing so, borders on misleading uninformed readers.

Minor comments
Abstract – the authors give nice information on protected areas in the four countries, but fail to state that many of these areas are ineffective at protecting primates, thus the statement borders on misleading.

Lines 250 and thereabouts: In the section of expanding agriculture, the authors really need to mention that growing human population and consumption patterns favoring meat are driving this expansion. This is presented later, but a line to this effect would be useful here.

Line 325: A mention of the fact should be made that that where oil is found, these governments will likely exploit it even if the oil is found in protected areas. This will occur if there is good or poor governance.

Line 491: I think a mention should be made of the fact that the logistics of administering vaccinations to great apes is extremely challenging to say the least, particularly since these animals are hunted.

Reintroductions: I think that the authors should mention the costs of these reintroductions. In a world where conservation funds are unlimited, this could be a priority, but since more primate populations are unprotected and are declining, these efforts should be seen as a costly option, relative to other options.

Conclusions and Key Challenges Ahead: I think that this section could be reduced and repetition to what has already been stated could be eliminated so that this section could have a stronger impact.

Experimental design

Not Applicable

Validity of the findings

Great

Additional comments

See above

Reviewer 2 ·

Basic reporting

No comment.

Experimental design

No comment.

Validity of the findings

No comment.

Additional comments

The manuscript entitled “Primates in peril: the significance of Brazil, Madagascar, Indonesia and the Democratic Republic of the Congo for global primate conservation” by Estrada et al. provides a comprehensive review of primate populations in these countries, as well as threats and anthropogenic pressures that these populations are facing. The authors also predict how the distribution would be affected by agriculture expansion in the future under business-as-usual scenario. The manuscript is generally well-written and could contribute to understanding of primate populations and threats to their survivals. I have some recommendations to improve the current presentation of the manuscript.

Line 96-97: I am not sure if the effectiveness of governance is really asses in the manuscript, as it would require a thorough causal analysis (with counterfactuals) to determine whether the current primate situation is attributed by good/poor governance (see Baylis et al. 2016). I would suggest moderating the language in the link between governance and conservation efforts.

Line 182-253: The section on “Large-scale encroachment and loss of primate habitats” perhaps can be divided into two sub-headings: (1) historical and (2) projected. The historical component would encompass the current “Forest loss” and “Expansion of agricultural land”. The projected component would encompass the “Modelling agricultural expansion and primate range contraction in the 21st century”.

Line 521-549: The impact of climate change on forest and primate populations through wildfire is not well mentioned. Mitigating climate change impact on fire is critical for the survival of primates in Indonesia (especially in Borneo and Sumatra where large areas of peatland have been degraded and vulnerable to fire, which could spread to adjacent forest).

Line 602-608: It may worth to assess how the Gini index (disparity in wealth) in each country based on the UN data. In some countries, the Gini index might have increased over time, which indicates increased disparity between the rich and the poor. So the average HDI index masks the actual distribution of poverty.

Line 745-776: There has been some recent progress on research in community forestry which can be worth to discuss in this section. Examples include: Rasolofoson et al. (2015, 2017) that shows how community forest management in Madagascar have successfully reduced deforestation and also improved local community wellbeing, Santika et al. (2017) that shows the impact of community forestry on reducing deforestation in Indonesia, and other studies from South America. Perhaps could also look at Bowler et al. (2012) and Burivalova et al. (2017) for recent review of community forestry and reduced impact logging and forest certification scheme.

References:

Baylis, K., Honey‐Rosés, J., Börner, J., Corbera, E., Ezzine‐de‐Blas, D., Ferraro, P.J., Lapeyre, R., Persson, U.M., Pfaff, A. and Wunder, S., 2016. Mainstreaming impact evaluation in nature conservation. Conservation Letters, 9(1), pp.58-64.

Bowler, D.E., Buyung-Ali, L.M., Healey, J.R., Jones, J.P., Knight, T.M. and Pullin, A.S., 2012. Does community forest management provide global environmental benefits and improve local welfare?. Frontiers in Ecology and the Environment, 10(1), pp.29-36.

Burivalova, Z., Hua, F., Koh, L.P., Garcia, C. and Putz, F., 2017. A critical comparison of conventional, certified, and community management of tropical forests for timber in terms of environmental, economic, and social variables. Conservation Letters, 10(1), pp.4-14.

Rasolofoson, R.A., Ferraro, P.J., Jenkins, C.N. and Jones, J.P., 2015. Effectiveness of community forest management at reducing deforestation in Madagascar. Biological Conservation, 184, pp.271-277.

Rasolofoson, R.A., Ferraro, P.J., Ruta, G., Rasamoelina, M.S., Randriankolona, P.L., Larsen, H.O. and Jones, J.P., 2017. Impacts of Community Forest Management on Human Economic Well‐Being across Madagascar. Conservation Letters, 10(3), pp.346-353.

Santika, T., Meijaard, E., Budiharta, S., Law, E.A., Kusworo, A., Hutabarat, J.A., Indrawan, T.P., Struebig, M., Raharjo, S., Huda, I. and Ekaputri, A.D., 2017. Community forest management in Indonesia: Avoided deforestation in the context of anthropogenic and climate complexities. Global Environmental Change, 46, pp.60-71.

---

## Round 0.2 · accepted · Accept

Thank you very much for your revised piece. The concerns of reviewers have been adequately met or addressed in the rebuttal. The additional information in text and as supplemental material meets the request to take a more critical approach to the mined data.

#